# ParFam – Symbolic Regression Based on Continuous Global Optimization

## Abstract

The problem of symbolic regression (SR) arises in many different applications, such as identifying physical laws or deriving mathematical equations describing the behavior of financial markets from given data. Various methods exist to address the problem of SR, often based on genetic programming. However, these methods are usually quite complicated and require a lot of hyperparameter tuning and computational resources. In this paper, we present our new method *ParFam* that utilizes parametric families of suitable symbolic functions to translate the discrete symbolic regression problem into a continuous one, resulting in a more straightforward setup compared to current state-of-the-art methods. In combination with a powerful global optimizer, this approach results in an effective method to tackle the problem of SR. Furthermore, it can be easily extended to more advanced algorithms, e.g., by adding a deep neural network to find good-fitting parametric families. We prove the performance of ParFam with extensive numerical experiments based on the common SR benchmark suit SRBench, showing that we achieve state-of-the-art results. Our code and results can be found at https://anonymous.4open.science/r/parfam-90FC.

## 1 Introduction

*Symbolic regression* (SR) describes the task of finding a symbolic function that accurately represents the connection between given input and output data. At the same time, the function should be as simple as possible to ensure robustness against noise and interpretability. This is of particular interest for applications where the aim is to (mathematically) analyze the resulting function afterward or get further insights into the process to ensure trustworthiness, for instance, in physical or chemical sciences (Quade et al., 2016; Angelis et al., 2023; Wang et al., 2019). The range of possible applications of SR is therefore vast, from predicting the dynamics of ecosystems (Chen et al., 2019), forecasting the solar power for energy production (Quade et al., 2016), estimating the development of financial markets (Liu & Guo, 2023), analyzing the stability of certain materials (He & Zhang, 2021) to planning optimal trajectories for robots (Oplatkova & Zelinka, 2007), to name but a few. Moreover, as Angelis et al. (2023) point out, the number of papers on SR has increased significantly in recent years, highlighting the relevance and research interest in this area.

SR is a specific regression task in machine learning that aims to find an accurate model without any assumption by the user related to the specific data set. Formally, a symbolic function $f : \mathbb{R}^n \to \mathbb{R}$ that accurately fits a given data set $(x_i, y_i)_{i=1,\dots,N} \subseteq \mathbb{R}^n \times \mathbb{R}$ is sought, i.e., it should satisfy $y_i = f(x_i)$ for all data points, or in the case of noise $y_i \approx f(x_i)$ for all $i \in \{1, \dots, N\}$. ~~Since, in this general setting, there are no assumptions on the structure of possible models, the search space is infinite-dimensional. In practice, however, it is necessary to specify the model space in some sense, and all methods rely in one way or another on certain implicit assumptions in the modeling process. For example, genetic programming (GP) methods, one of the most common classes of solution algorithms, require the choice of base functions that can be combined to build the model.~~ Unlike other regression tasks, SR aims at finding a simple symbolic and thus interpretable formula while assuming as little as possible about the unknown function. In contrast to SR, solutions derived via *neural networks* (NNs), for instance, lack interpretability and traditional regression tasks typically assume a strong structure of the unknown function like linearity or polynomial behavior.

To tackle SR problems, the most established methods are based on *genetic programming* (Augusto & Barbosa, 2000; Schmidt & Lipson, 2009; 2010; Cranmer, 2023), but nowadays there also exist many solution algorithms that make use of other machine learning methods, in particular neural networks (Udrescu & Tegmark, 2020; Martius & Lampert, 2017; Desai & Strachan, 2021; Makke et al., 2022). However, even though there have been many attempts with complicated procedures to search through the infinite-dimensional space of functions, many of them show unsatisfactory results when evaluated on complex benchmarks: La Cava et al. (2021) evaluate 13 state-of-the-art SR algorithms on the *SRBench* ground-truth problems: the Feynman (Udrescu & Tegmark, 2020) and Strogatz (La Cava et al., 2016) problem sets. Both data sets consist of physical formulas with varying complexities, where the first one encompasses 115 formulas and the latter 14 ordinary differential equations. Out of the 13 algorithms evaluated by La Cava et al. (2021), all algorithms find at most 30% of the formulas of each problem set in the given time of 8 CPU hours, except for AI Feynman (Udrescu & Tegmark, 2020). AI Feynman, which is based on recursive function simplification inspired by the structure of the Feynman data set, is able to recover more than 50% of the Feynman equations but fails for more than 70% for the Strogatz problems. The rates are even worse for data sets incorporating noise (La Cava et al., 2021; Cranmer, 2023). In addition to AI Feynman, we are only aware of one other algorithm, proposed by Holt et al. (2023) after the benchmark by La Cava et al. (2021), which has demonstrated superior performance on the SRBench ground-truth data sets while following the SRBench time limit.

In this paper, we introduce the novel algorithm *ParFam* that addresses SR by leveraging the inherent structure of physical formulas and, thereby, translating the discrete optimization problem into a continuous one. This grants users precise control over the search space and facilitates the incorporation of gradient-based optimization techniques. More precisely, we apply *basin-hopping*, which combines a global random search with a local search algorithm (Wales & Doye, 1997). Originally, this algorithm was designed to solve molecular problems and, thus, is suitable for very high-dimensional landscapes. The details of ParFam are introduced in Section 2.1. Notably, despite its straightforward nature, ParFam achieves state-of-the-art performance on the Feynman and Strogatz data set as demonstrated in Section 3.1. Moreover, this structure enables the simple application of pre-trained NNs to reduce the dimensionality of the search space. This concept is exemplified by our prototype, *DL-ParFam*, introduced in Section 2.2. Through experimentation on a synthetic data set, we demonstrate that DL-ParFam significantly surpasses ParFam, cf. Section 3.2.

**Our Contributions**   Our key contributions are as follows:

1. We introduce ParFam, a new method for SR leveraging the inherent structure of physical formulas and, thereby, translating the discrete optimization problem into a continuous one. This results in the following advantages: (1) Enabling gradient-based optimization techniques; (2) Efficient but simple and user-friendly setup; (3) State-of-the-art performance on the ground-truth problems of La Cava et al. (2021), the Feynman and Strogatz data sets.

2. Furthermore, we introduce a prototype of the extension DL-ParFam, which shows how the structure of ParFam allows for using a pre-trained NN, potentially overcoming the limitations of previous approaches.

**Related work**   Traditionally, genetic programming have been used for SR to heuristically search the space of equations given some base functions and operations (Augusto & Barbosa, 2000; Schmidt & Lipson, 2009; 2010; Cranmer, 2023). However, due to the accomplishments of neural networks across diverse domains, numerous researchers aimed to leverage their capabilities within the realm of SR. Udrescu & Tegmark (2020), for instance, have employed an auxiliary NN to evaluate data characteristics. In a similar vein, Martius & Lampert (2017), Sahoo et al. (2018), and Desai & Strachan (2021) used compact NN architectures with physically meaningful activation functions, such as $\sin$ and $\cos$, enabling stochastic gradient descent to search for symbolic functions.

The approach by Petersen et al. (2021), on the contrary, relies on *reinforcement learning* (RL) to explore the function space, where a policy, modeled by a recurrent neural network, generates candidate solutions. Mundhenk et al. (2021) combined this concept with genetic programming such that the RL algorithm iteratively learns to identify a good initial population for the GP algorithm, resulting in superior performance compared to individual RL and GP approaches. Similarly, Sun et al. (2022) rely on Monte Carlo tree search to search the space of expression trees for the correct equations.

Given the simplicity of sampling functions and evaluating them, several endeavors have emerged to train NNs using synthetic data to predict underlying functions. Initial simpler approaches were limited by the variability of the given data set (Biggio et al., 2020) or of the functions (Li et al., 2022). However, these limitations can be effectively circumvented by more advanced approaches using the transformer architecture (Biggio et al., 2021; Kamienny et al., 2022; Holt et al., 2023).

Apart from the algorithms evaluated by La Cava et al. (2021), Deep Generative Symbolic Regression (DGSR) by Holt et al. (2023) and unified Deep Symbolic Regression (uDSR) by Landajuela et al. (2022) are the only algorithms—to the best of our knowledge—which have been evaluated on the whole Feynman data set and outperformed the state-of-the-art AI Feynman in the symbolic recovery rate. Notably, DGSR's success has only been possible by the computationally expensive pre-training of an encoder-decoder architecture using RL instead of gradient-based methods to learn invariances of the functions and an additional finetuning step during inference by performing neural guided priority queue training (NGPQT) as introduced by Mundhenk et al. (2021). uDSR builds upon the already well-performing AI Feynman and adds pre-training, genetic programming, reinforcement learning, and linear regression to it. Unlike for the SRBench benchmark (La Cava et al., 2021), Landajuela et al. (2022) evaluate their method with a time limit of 24 instead of 8 hours, which is why we omit uDSR from our comparisons in Section 3.

Most SR algorithms approach the problem by first searching for the analytic form of $f$ and then optimizing the resulting coefficients. In contrast, only a few algorithms follow the same idea as ParFam, to merge these steps into one by spanning the search space using an expressive parametric model and searching for sparse coefficients that simultaneously yield the analytical function and its coefficients. FFX (McConaghy, 2011) and SINDy (Brunton et al., 2016) utilize a model to span the search space which is linear in its parameters, to be able to apply efficient methods from linear regression to compute the coefficients. To increase the search space, they construct a large set of features by applying the base functions to the input variables. While these linear approaches enable fast processing in high dimensions, they are unable to model non-linear parameters within the base functions, restricting the search space to a predefined set of features.

The closest method to ParFam is EQL (Martius & Lampert, 2017; Sahoo et al., 2018), which overcomes this limitation by utilizing small neural networks with $\sin$, $\cos$, and the multiplication as activation function. The goal of EQL is to find sparse weights, such that the neural network reduces to an interpretable formula. However, while EQL applies linear layers between the base functions, ParFam applies rational layers. Thereby, ParFam is able to represent most relevant functions with its one layer of base functions, while EQL usually needs multiple ones, which introduces many redundancies, inflates the number of parameters, and complicates the optimization process. ~~ParFam shares conceptual proximity with EQL, as both methods assume a structure of general formulas, effectively translating SR into a continuous optimization problem. However, while ParFam aims to guarantee a unique parameterization for each function, EQL exhibits many redundancies that inflate the parameter space.~~ Moreover, EQL relies on the local minimizer ADAM (Kingma & Ba, 2014) for coefficient optimization. On the contrary, ParFam leverages the reduced dimensionality of the parameter space by applying global optimization techniques for the parameter search, which mitigates the issues of local minima. Furthermore, ParFam maintains versatility, allowing for the straightforward inclusion of the base functions, while EQL cannot handle the exponential, logarithm, root, and division within unary operators. Similar to DL-Parfam, Liu et al. (2023) enhanced EQL with a pre-training step. However, this approach still suffers from the listed structural limitations of EQL.

## 2 METHODS

In the following section, we first introduce our new method ParFam, that exploits a well-suited representation of possible symbolic functions to which an efficient global optimizer can be applied. Afterward, we discuss the extension DL-ParFam, which aims to enhance ParFam by utilizing deep learning to obtain better function representations.

### 2.1 PARFAM

The aim of SR is to find a simple and thus interpretable function that describes the mapping underlying the data $(x_i, y_i)_{i=1,...,N}$ without many additional assumptions. Typically, a set of base functions,

such as $\{+, -, ^{-1}, \exp, \sin, \sqrt{}\}$, is predetermined. The primary goal of an SR algorithm is to find the simplest function that uses only these base functions to represent the data, where simplicity is usually defined as the number of operations. Since most algorithms make no other assumptions on the function they are looking for, this approach results in a search space that grows exponentially in the number of base functions, dimensions of $x$, and depth of the expression trees.

To reduce the complexity of the search space on the one hand and to obtain more meaningful results on the other hand, some methods apply filters to prevent the output of unwanted or "unnatural" functions. For instance, Petersen et al. (2021) prevent their algorithm from creating compositions of trigonometric functions as $\sin \circ \cos$ since these are rarely encountered in any scientific domain. Given that the main idea of SR is to gain knowledge of scientific processes, such structural assumptions appear to be reasonable. This is also the motivation for restricting the search space in our approach. Furthermore, we choose the function space such that it can be represented by a parametric family, and the proper expression can be found by applying a continuous global optimizer.

### 2.1.1 THE STRUCTURE OF THE PARAMETRIC FAMILY

The main motivation for ParFam is that most functions appearing in real-world applications can be represented by functions of certain parametric families. More precisely, we assume that they can be written in the form

$$f_\theta(x) = Q_{k+1}(x, g_1(Q_1(x)), g_2(Q_2(x)), \dots, g_k(Q_k(x))), \tag{1}$$

where $Q_1, ..., Q_{k+1}$ are rational functions, $g_1, ..., g_k$ are the unary base functions, which cannot be expressed as rational functions, like $\sin$, $\exp$, $\sqrt{}$ etc., and $x \in \mathbb{R}^n$ is the input vector. Moreover, $\theta \in \mathbb{R}^m$ denotes the coefficients of the individual polynomials, i.e., of the numerators and denominators of $Q_1, ..., Q_{k+1}$, which are the learnable parameters of this family of functions. The degrees $d_i^1$ and $d_i^2$, $i \in \{1, \dots, k+1\}$, of the numerator and denominator polynomials of $Q_1, ..., Q_{k+1}$, respectively, and the base functions $g_1, ..., g_k$ are chosen by the user. Depending on the application, even specialized custom functions can be added to the set of base functions. This versatility and its simplicity make ParFam a highly user-friendly tool, adaptable to a wide range of problem domains. In Appendix A, we explain how to incorporate specific base functions to avoid numerical issues and further implementation details.

The parametric family we consider excludes composite functions such as $\sin \circ \cos$ similarly to Petersen et al. (2021). This is rooted in the structure of physical formulas we observe, as can be seen, e.g., in the set of ground-truth problems from SRBench (La Cava et al., 2021), which consists of 129 physically meaningful formulas and only includes one function that does not follow equation 1:

$$\sqrt{(x_1^2 + x_2^2 - 2x_1 x_2 \cos(\theta_1 - \theta_2))} \text{ (Udrescu \& Tegmark, 2020, I.29.16).}$$

Furthermore, the "Cambridge Handbook of Physics Formulas" (Woan, 2000) contains more than 2,000 equations from the major physics topics, among which only a handful do not follow the structure of equation 1. In addition, the structure of ParFam is chosen due to its inherent interpretability, avoiding complicated compositions, and its high expressivity even if the true formula cannot be recovered, as shown by our experiments in Section 3.1.

### 2.1.2 OPTIMIZATION

Restricting the search space to functions of the parametric family given by equation 1 yields the advantage that we can translate the discrete SR problem into a continuous one, as now the task is to find the parameters of the rational functions $Q_1, ..., Q_{k+1}$ such that $f_\theta$ approximates the given data $(x_i, y_i)_{i=1,...,N}$, i.e., we aim to minimize the average $l_2$-distance between $y_i$ and $f_\theta(x_i)$. As we aim for preferably simple functions to derive interpretable and easy-to-analyze results, a regularization term $R(\theta)$ is added to encourage sparse parameters. In total, we aim at minimizing the loss function

$$L(\theta) = \frac{1}{N} \sum_{i=1}^{N} (y_i - f_\theta(x_i))^2 + \lambda R(\theta), \tag{2}$$

where $\lambda > 0$ is a hyperparameter to control the weight of the regularization. Here, we choose $R(\theta) = \|\theta\|_1$ as a surrogate for the number of non-zero parameters, which is known to enforce sparsity in other areas, e.g., NN training (Bishop, 2006; Goodfellow et al., 2016). In Appendix A, we discuss how to deal with the regularization of the coefficients of rational functions in detail.

Although the SR problem is now transformed into a continuous optimization problem, due to the presence of many local minima, it is not sufficient to apply purely local optimization algorithms like gradient descent or BFGS (Nocedal & Wright, 2006). This is also ~~discussed by Nocedal & Wright (2006) and~~ shown in our comparison study in Appendix B. To overcome these local minima, we instead rely on established (stochastic) global optimization methods. Here, we choose the so-called *basin-hopping* algorithm, originally introduced by Wales & Doye (1997), which combines a local minimizer, e.g., BFGS (Nocedal & Wright, 2006), with a global search technique inspired by Monte-Carlo minimization as proposed by Li & Scheraga (1987) to cover a larger part of the parameter space. More precisely, we use the implementation provided by the SciPy library (Virtanen et al., 2020). Each iteration consists of three steps:

1. Random perturbation of the parameters.
2. Local minimization, e.g., with the BFGS method.
3. Acceptance test based on the function value of the local optimum.

The basic idea of the algorithm is to divide the complex landscape of the loss function into multiple areas, leading to different optima. These are the so-called basins. The random perturbation of the parameters allows for hopping between these basins and the local search (based on the real loss function) inbetween improves the results and ensures that a global minimum is reached if the correct basin is chosen. For the acceptance test, the criterion introduced by Metropolis et al. (1953) is taken.

Following the optimization with basin-hopping, a finetuning routine is initiated. In this process, coefficients that fall below a certain threshold are set to 0, and the remaining coefficients are optimized using the L-BFGS method, starting from the previously found parameters. The threshold is gradually increased from $10^{-5}$ to $10^{-2}$ to encourage further sparsity in the discovered solutions. This step has been found to be crucial in enhancing the parameters initially found by basin-hopping.

## 2.2 DL-ParFam

As discussed in the related work section, there have been multiple attempts in recent years to leverage pre-training for SR, as synthetic data can be easily generated. Even though modern approaches are able to handle flexible data sets in high dimensions (Biggio et al., 2021; Kamienny et al., 2022), they fail to incorporate invariances in the function space during training, e.g., $x + y$ and $y + x$ are seen as different functions, as pointed out by Holt et al. (2023), which possibly complicates the training. Holt et al. resolve this by evaluating the generated function to compute the loss and update the network using RL. This effectively solves the invariance problem, as can be seen by their state-of-the-art symbolic recovery rate on the Feynman data set. However, evaluating each function during the training instead of comparing its symbolic expression with the ground-truth is computationally expensive. Moreover, due to the non-differentiability, the network has to be optimized using suitable algorithms like Policy gradient methods.

Here, we propose our approach DL-ParFam, which aims to combine the best of both worlds. The idea is to use an NN that, given a data set $(x_i, y_i = f(x_i))_{i=1,...,N}$, predicts a sparsity pattern on the coefficients $\theta$ of the parametric family in equation 1. This sparsity pattern or mask specifies which parameters should be variable and learned in the ParFam algorithm and which can be ignored and set to a fixed value of 0. The idea of DL-ParFam is visualized in Figure 1. This approach yields significant improvements compared to ParFam and other pre-training based SR methods:

- Compared to ParFam: DL-ParFam strongly reduces the dimensions of the optimization problem considered in ParFam, effectively reducing the computation time and success rate for any global optimizer.
- Compared to other pre-training based SR methods: DL-ParFam predicts the structure of the function, which can be directly compared with the ground-truth and, thereby, avoids the evaluation on the data grid in every training step and yields an end-to-end differentiable pipeline. In addition, DL-ParFam adeptly handles function invariances, as we guarantee that each set of parameters uniquely defines a function via the structure of ParFam.

The primary purpose of this subsection is to demonstrate the potential of utilizing ParFam as a means of structuring scientific formulas beyond the direct optimization presented so far. Our intent is not

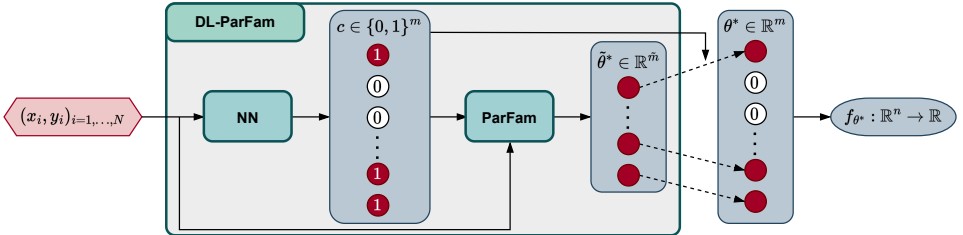

Figure 1: Schematic illustration of the DL-ParFam method: DL-ParFam uses the data $(x_i, y_i)_{i=1,\ldots,N}$ as the input of a pre-trained neural network which predicts a mask $c \in \{0,1\}^m$ in the parameter space. As usual, ParFam then aims to find parameters $\theta \in \mathbb{R}^m$ to minimize the loss as defined in equation 2. However, instead of optimizing each entry of $\theta$, ParFam only optimizes those parameters $\theta_k$ for which $c_k = 1$ and keeps the others as 0, essentially reducing the parameter space to $\tilde{m} = \sum_{k=1}^{m} c_j$.

to present an implementation of DL-ParFam in this paper that can rival existing deep learning-based methods on complex benchmarks like the Feynman data set. This decision is driven by the myriad challenges inherent in benchmarking, such as high dimensionality, diverse data point distributions, varying numbers of data points and dimensions, and a plethora of base functions. While these challenges can be addressed using deep learning techniques, as demonstrated by Biggio et al. (2021), Kamienny et al. (2022), and Holt et al. (2023), they require specialized architectures. Since the main focus of this paper is ParFam and the particular choices are not directly related to ParFam, we opt for a straightforward implementation of DL-ParFam to demonstrate its effectiveness on synthetic data.

The vanilla implementation, which we consider here, uses a simple fully-connected feedforward neural network $NN : \mathbb{R}^N \to \mathbb{R}^m$ which takes as input the data $(y_i)_{i=1,\ldots,N}$ and outputs a mask $c \in [0,1]^m$, where $c_i$ represents the likelihood that $\theta_i$ is needed to represent the sought symbolic function, i.e., $c_i \approx 0$ indicates that the parameter $\theta_i$ is supposed to be 0 and $c_i \approx 1$ indicates $\theta_i \neq 0$. To reduce the dimensionality of the NN, we only take the output data $y$ of the functions as input to the NN. Thus, we implicitly assume that the input data is sampled on the same grid $(x_i)_{i=1,\ldots,N} \subseteq \mathbb{R}^n$ for all data sets. To ensure this, we train the NN on synthetically generated data $(y^j = (y_i^j)_{i=1,\ldots,N}, c^j)_{j=1,\ldots,K}$, where $c^j \in [0,1]^m$ is some mask and $y_i^j = f_{\theta^j}(x_i)$ is the output of the corresponding function evaluated on the fixed grid point $x_i \in \mathbb{R}^n$. As a loss function with respect to the parameters $w$ of the NN, we define

$$L_{\mathsf{NN}}(w) = \sum_{j=1}^{K} \mathrm{BCE}(NN(y^j), c^j), \tag{3}$$

where $\mathrm{BCE} : [0,1]^m \times [0,1]^m \to \mathbb{R}$ denotes the binary-entropy loss, i.e.,

$$\mathrm{BCE}(\bar{c}, c) = \frac{1}{K} \sum_{l=1}^{m} - \left( c_l \log(\bar{c}_l) + (1 - c_l) \log(1 - \bar{c}_l) \right). \tag{4}$$

Another important difference from previous approaches, not outlined before, is that DL-ParFam combines the experience gained through pre-training with the power of the method ParFam, which is highly competitive on its own. In Section 3.2, we show the value of this combination.

## 3 BENCHMARK

In Section 3.1, we evaluate ParFam on the Feynman (Udrescu & Tegmark, 2020) and Strogatz (La Cava et al., 2016) data sets and report its performance in terms of the symbolic recovery rate, the coefficient of determination $R^2$, and the complexity of the derived formula showing that our simple setup significantly outperforms most existing methods and reaches state-of-the-art in SR. In Section 3.2, we study a prototype of DL-ParFam, revealing the vast potential of adding pre-training to ParFam.

### 3.1 PARFAM

After the introduction of the SR benchmark (SRBench) by La Cava et al. (2021), several researchers, including Mundhenk et al. (2021), Holt et al. (2023), Kamienny et al. (2022), and Biggio et al.

(2021), have reported their findings using the SRBench's ground-truth data sets. These data sets are the Feynman and the Strogatz data set.

**Feynman data set** The Feynman data set consists of 119 physical formulas taken from the Feynman lectures and other seminal physics books (Udrescu & Tegmark, 2020). Some examples can be found in Appendix C. The formulas depend on a maximum of 9 independent variables and are composed of the elementary functions $+, -, *, /, \sqrt{}, \exp, \log, \sin, \cos, \tanh, \arcsin$ and $\arccos$. Following La Cava et al. (2021), we omit three formulas containing $\arcsin$ and $\arccos$ and one data set where the ground-truth formula is missing. Additionally, since the data sets contain more data points than required for ParFam and this abundance of data slows down the optimizer, we only consider a subset of 500, for the experiments without noise, and 1,000, for the experiments with noise, data points of the training data for each problem.

**Strogatz data set** The Strogatz data set introduced by La Cava et al. (2016) is the second ground-truth problem set included in SRBench (La Cava et al., 2021). It consists of 14 non-linear differential equations describing seven chaotic dynamic systems in two dimensions, listed in Appendix D. Each data set contains 400 samples.

**Metrics** To ensure comparability with the results evaluated on SRBench, we use the same evaluation metrics as La Cava et al. (2021). First, we report the symbolic recovery rate, which is the percentage of equations ParFam recovered. Second, we consider the coefficient of determination

$$R^2 = 1 - \frac{\sum_{i=1}^{N}(y_i - \hat{y}_i)^2}{\sum_{i=1}^{N}(y_i - \bar{y})^2}, \tag{5}$$

where $\hat{y}_i = f_\theta(x_i)$ represents the model's prediction and $\bar{y}$ the mean of the output data $y$. The closer $R^2$ is to 1, the better the model describes the variation in the data. It is a widely used measure for goodness-of-fit since it is independent of the scale and variation of the data. Lastly, we report the complexity of our formula based on the number of mathematical operations following the definition in SRBench. The original data sets do not include any noise. However, similar to La Cava et al. (2021), we additionally perform experiments with noise by adding $\epsilon_i \sim N\left(0, \sigma^2 \frac{1}{N} \sum_{i=1}^{N} y_i^2\right)$ to the targets $y_i$, where $\sigma$ denotes the noise level.

**Hyperparameters** The hyperparameters of ParFam can be divided into two subsets. The first subset defines the parametric family $(f_\theta)_{\theta \in \mathbb{R}^m}$, e.g., the degree of the polynomials and the set of base functions. A good choice for this set is highly problem-dependent. However, in the absence of prior knowledge, it is advantageous to select a parametric family that is sufficiently expansive to encompass a wide range of potential functions. In this context, we opt for $\sin, \exp,$ and $\sqrt{}$ as our base functions. For the "input rational functions" $Q_1, \ldots, Q_k$, we set the degrees of the numerator and denominator polynomials to 2. For $Q_{k+1}$, we set the degree of the numerator polynomial to 4 and the denominator polynomial to 3. This choice results in a parametric family with hundreds of parameters, making it challenging for global optimization. To address this issue, we iterate through various smaller parametric families, each contained in this larger family, see Appendix E for details. The second set of hyperparameters defines the optimization scheme. Here, we set the regularization parameter to $\lambda = 0.001$, the number of iterations for basin-hopping to 10, and the maximal number of BFGS steps for the local search to 100 times the dimension of the problem. Our choice of parameters is summarized in Table 4 in Appendix F.

**Results** Following La Cava et al. (2021), we allow a maximal training time of 8 CPU hours and a maximal number of function evaluations of $1,000,000$. In Figure 2a, we present the symbolic recovery rate on both data sets together. ParFam, AI Feynman, and DGSR exhibit exceptional performance, outperforming all other competitors by a substantial margin (over $25\%$). It is important to note that AI Feynman performs particularly well on the Feynman data sets but fails on the Strogatz data set, as shown in Appendix G, indicating that the algorithm is tailored to the Feynman data set. Since DGSR was not tested on noisy data and AI Feynman is strongly influenced by noise, ParFam outperforms all competitors at a noise level of $\sigma = 0.01$. Furthermore, Figure 2b shows the accuracy solution, which is the percentage of formulas for which $R^2 > 0.999$ holds on the test sets. Here, ParFam outperforms all competitors with and without noise. It is important to note that Holt et al.

(2023) reported a similar metric but with $R^2 > 0.99$ instead of $R^2 > 0.999$. However, DGSR achieved a value of $90.95\%$ for this less strict metric, compared to ParFam's $97.67\%$. Figure 3 reveals that ParFam achieves a mean $R^2$ significantly better than its competitors, albeit with slightly more complex formulas. Note that the mean, rather than the median, is shown in this figure since both ParFam and AI Feynman solve over 50% of the formulas without noise, causing their median performance to simply reflect this high success rate. The corresponding plot showing the median and additional results can be found in Appendix G. We performed the experiments without tuning the hyperparameter $\lambda$. To assess the sensitivity of the results with respect to $\lambda$, see Appendix H.

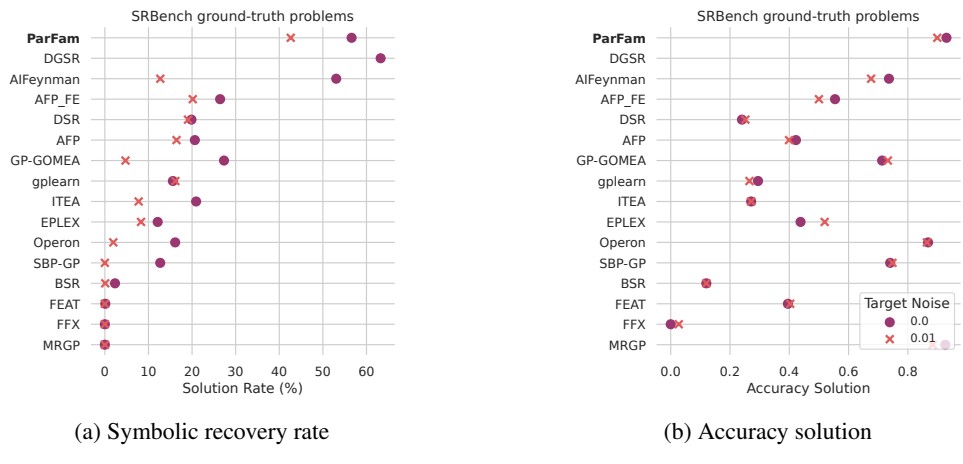

(a) Symbolic recovery rate          (b) Accuracy solution

Figure 2: Symbolic recovery and accuracy solution rate (percentage of data sets with $R^2 > 0.999$ for the test set) on the SRBench ground-truth problems (Feynman and Strogatz data sets).

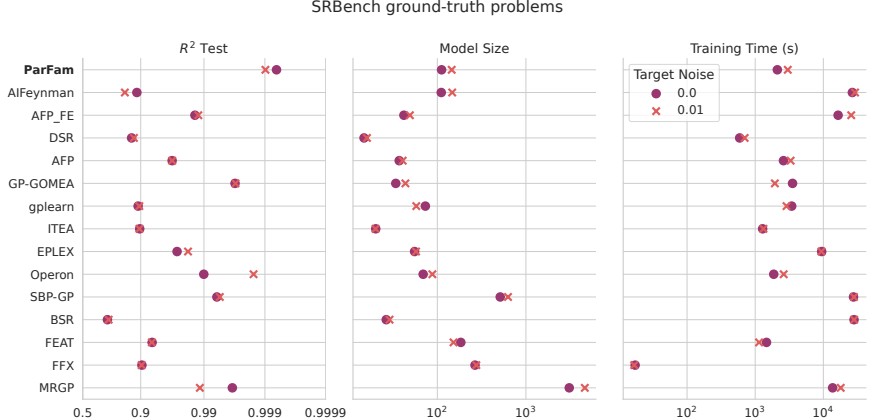

Figure 3: Mean results on the SRBench ground-truth problems.

Due to the proximity of EQL (Martius & Lampert, 2017; Sahoo et al., 2018) and ParFam, we deem a comparison between these two methods as highly interesting, however, the restricted expressivity of EQL makes it an unfair comparison on the whole Feynman and Strogatz dataset. For this reason, we show the results for EQL on a reduced benchmark in Appendix I. For results of ParFam on the Nguyen benchmark (Uy et al., 2011) and comparisons with algorithms that were not tested on SRBench, like SPL (Sun et al., 2022) and NGGP (Mundhenk et al., 2021), see Appendix J.

## 3.2 DL-PARFAM

~~In this subsection, we aim to demonstrate the potential of DL-ParFam by conducting experiments on synthetically generated data sets.~~ Due to the prototype status of DL-ParFam, the ability to evaluate it on complex data sets, such as the Feynman dataset, is limited as the data to be processed is not sampled on the same grid. Therefore, we use synthetic data sets.

Table 1: Relative performance and runtime of ParFam and DL-ParFam on the synthetic data set

| | ParFam | | DL-ParFam | |
|---|---|---|---|---|
| # Iterations | Symbolic recovery | Training time | Symbolic recovery | Training time |
| 50 | 32% | 13s | 48% | 7s |
| 100 | 30% | 24s | 46% | 12s |
| 500 | 43% | 116s | 66% | 55s |
| 1000 | 37% | 221s | 69% | 107s |

**Synthetic data sets**  To generate the synthetic data sets, we fix the grid $x_i = -10 + 0.1i$, $i \in \{1, ..., N = 200\}$, and choose one set of model hyperparameters to define a specific parametric family $(f_\theta)_{\theta \in \mathbb{R}^m}$. Here, we choose the base functions $\sin$ and $\sqrt{}$, set the degree of all numerator polynomials to 2 and of all denominator polynomials to 0. Then, we sample $(\theta^j)_{j \in \{1,...,K\}} \subset \mathbb{R}^m$ following the scheme described in Appendix K. For each $\theta^j$, $j \in \{1, \ldots, K\}$, and each $x_i$ we evaluate $f_{\theta^j}(x_i) = y_i^j$ to obtain $K$ data sets $((x_i)_{i=1,...,N}, (y_i^j)_{i=1,...,N}, \theta^j)_{j=1,...,K}$. For our numerical tests, we create two different data sets: The first includes 2,000,000 equations for training the neural network of DL-ParFam, 10,000 for its validation, and another 10,000 for its testing. The second set consists of 100 formulas to compare DL-ParFam with ParFam. Our hyperparameter choices are summarized in Table 10 in Appendix L.

**Pre-training of DL-ParFam**  We construct the neural network of DL-ParFam as a feedforward NN with one hidden layer containing 200 neurons. It is trained as described in Section 2.2 using the Adam optimizer with a learning rate of 0.0001 and 20,000 epochs with 500 batches, which takes less than 4h on a TITANRTX GPU. In addition, to predict a mask $c \in \{0,1\}^m$ as described in Section 2.2 we set $c_k = 0$ if $NN((y_i)_{i=1,...,N}) < 0.2$ and $c_k = 1$ otherwise.

**Metrics**  To evaluate the performance of the NN, we report two metrics: the covering score and the average successful cover size. The covering score describes the percentage of formulas for which the mask includes the non-zero parameters, i.e., $\theta_k^j \neq 0$ implies $c_k^j = 1$. The average successful cover size is the mean over the means of $c^j$ across all formulas for which the NN succeeded at identifying the non-zero parameters. Ideally, this value should be as small as possible, indicating that the mask size is minimized while still effectively capturing the non-zero parameters. To understand the influence of the NN in DL-ParFam, we evaluate DL-ParFam and ParFam against each other on the second synthetic data set and report the symbolic recovery rate. Here, we assume for both methods that the perfect choice of model parameters is known, i.e., the same that were used to create the data sets. This assumption allows us to assess the relative performance of DL-ParFam and ParFam rather than evaluating their general performance.

**Results**  The NN reaches a covering score of $91.32\%$ on the test data set with an average successful cover size of $26.62\%$. This indicates that the pre-training helps to reduce the number of parameters by almost 3/4 in $91.32\%$ of the formulas. The relative performance of ParFam and DL-ParFam is shown in Table 1, which reveals that DL-ParFam solves consistently $16$-$32\%$ more equations than ParFam while requiring only approximately half the time.

## 4 DISCUSSION AND CONCLUSION

This work introduces ParFam along with its potential extension, DL-ParFam. Despite its inherent simplicity, ParFam demonstrates remarkable performance, as shown in Section 3.1. Furthermore, its adaptable structure makes it highly versatile for specific application scenarios. While DL-ParFam currently only exists in a prototype form, it already shows the feasibility and potential of integrating pre-training—a crucial direction in SR as pointed out by Kamienny et al. (2022); Biggio et al. (2021); Holt et al. (2023)—into the ParFam framework.

**Limitations**  While the structure of the parametric family of ParFam is undoubtedly its greatest asset in tackling SR, it can also be considered its most significant constraint, given that it imposes a

tighter constraint on the function space compared to other methods. However, Figure 2 illustrates, on the one hand, that several algorithms theoretically capable of identifying specific formulas do not always achieve this in practice. On the other hand, it demonstrates that even if ParFam restricts the function space too much, it still manages to find a formula that approximates the original function with remarkable accuracy. Another limitation of ParFam is that optimizing high-dimensional problems ($>10$ independent variables) is computationally expensive, given the exponential growth in the number of parameters with respect to the number of variables.

**Future Work** Subsequent efforts will concentrate on advancing both ParFam and DL-ParFam. With ParFam, several avenues remain unexplored, encompassing diverse forms of regularization, alternative parametrizations, and the potential incorporation of custom-tailored optimization techniques. Nonetheless, our primary focus will be on DL-ParFam, driven by its promising potential, as evidenced by our experiments. Numerous design choices await exploration, including data sampling strategies, choice of loss function, architecture selection, and more. Existing research in this direction will undoubtedly serve as valuable guidance (Kamienny et al., 2022; Biggio et al., 2021; Holt et al., 2023). We anticipate that these advancements will facilitate the deployment of even more expansive parametric families, thereby mitigating the limitations outlined earlier.

## REPRODUCIBILITY STATEMENTS

In our repository https://anonymous.4open.science/r/parfam-90FC, we include the results of our experiments and the code and instructions necessary to use our algorithms and reproduce all experiments shown in Section 3. Furthermore, we report all settings used in the experiments in the Appendices F and L.

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

# A    IMPLEMENTATION DETAILS

In this section, some further implementation details are discussed.

## A.1    REGULARIZATION OF THE DENOMINATOR

Since we aim for simple function representations, i.e., for sparse solutions $\theta \in \mathbb{R}^m$, the regularization term $R(\theta)$ is of great importance. If we parameterize a rational function $Q : \mathbb{R} \to \mathbb{R}$ in one dimension by

$$Q(x) = Q_{(a,b)}(x) = \frac{\sum_{i=0}^{d^1} a_i x^i}{\sum_{i=0}^{d^2} b_i x^i} \tag{6}$$

with $a \in \mathbb{R}^{d^1+1}$ and $b \in \mathbb{R}^{d^2+1}$, the following problem occurs: Since for any $\gamma \in \mathbb{R} \setminus \{0\}$ and $(a, b) \in \mathbb{R}^{d^1+1} \times \mathbb{R}^{d^2+1}$ it holds that $Q_{(a,b)}(x) = Q_{(\gamma a, \gamma b)}(x)$, the parameters cannot be uniquely determined. Although the non-uniqueness of the solution is not a problem in itself, it shows that this parameterization is not the most efficient, and, more importantly, the regularization will be bypassed since $\gamma$ can be chosen arbitrarily small. We address this issue by normalizing the coefficients of the denominator, i.e., we use $\tilde{b} = \frac{b}{||b||_2}$ rather than $b$. In other words, instead of defining rational functions by equation 6, we consider

$$Q_{(a,b)}(x) = \frac{\sum_{i=0}^{d^1} a_i x^i}{\frac{1}{||b||_2} \sum_{i=0}^{d^2} b_i x^i}. \tag{7}$$

Note that using the 2-norm and not the 1-norm is important since we regularize the coefficients using the 1-norm. To illustrate this, let $\tilde{b} = \frac{1}{||b||_p} b$.

Case $p = 1$: When $p = 1$, we have $||\tilde{b}||_1 = 1$ for any $b \in \mathbb{R}^{d^2}$. This demonstrates that $\tilde{b}$ is not regularized anymore and, consequently, also $b$ is not regularized. In essence, this choice of $p$ does not promote sparsity in the solution.

Case $p = 2$: In contrast, when $p = 2$, we have $||\tilde{b}||_1 = ||\frac{b}{||b||_2}||_1$. This expression favors sparse solutions, as it encourages the elements of $\tilde{b}$ to be close to zero, thus promoting regularization and sparsity in the solution.

## A.2    MISCELLANEOUS

In general, we look for rational functions $Q_i$ whose numerator and denominator polynomials have a degree greater than 1 in order to model functions like $x_1^2 \exp(2x_2)$. However, for some base functions, such as $\exp, \sqrt{}, \sin, \cos$, higher powers introduce redundancy, for instance, $\exp(x_2)^2 = \exp(2x_2)$. To keep the dimension of the parameter space as small as possible without limiting the expressivity of ParFam, we allow the user to specify the highest allowed power of each chosen base function separately. In our experiments, we set it to 1 for all used basis functions: $\exp, \cos$ and $\sqrt{}$.

To ensure that the functions generated during the optimization process are always well-defined and we do not run into an overflow, we employ various strategies:

- To ensure that $\sqrt{Q(x)}$ is well-defined, i.e., $Q(x) \geq 0$ for all $x$ in the data set, we instead use $\sqrt{|Q(x)|}$.
- To avoid the overflow that may be caused by the exponential function, we substitute it by the approximation $\min\{\exp(Q(x)), \exp(10) + |Q(x)|\}$, which keeps the interesting regime but does not run into numerical issues for big values of $Q(x)$. However, adding $|Q(x)|$ ensures that the gradient still points to a smaller $Q(x)$.
- To stabilize the division and avoid the division by 0 completely, we substitute the denominator by $10^{-5}$ if its absolute value is smaller than $10^{-5}$.

Implementing further base functions can be handled in a similar way as for the square root if they are only defined on a subset of $\mathbb{R}$ or are prone to cause numerical problems.

# B  OPTIMIZER COMPARISON

As discussed in the main paper, ParFam needs to be coupled with a powerful (global) optimizer to approximate the desired function. This section compares different global optimizers, underpinning our decision to use basin-hopping. We tested the following optimizers, covering different global optimizers and local optimizers combined with multi-start:

- L-BFGS with multi-start (Nocedal & Wright, 2006)
- BFGS with multi-start (Nocedal & Wright, 2006)
- Basin-hopping (Wales & Doye, 1997)
- Dual annealing (Xiang et al., 1997)
- Differential evolution (Wormington et al., 1999)

We conducted the experiments on a random subset of 15 Feynman problems, which are listed in Table 2 in Appendix C. For each of the 15 problems, we ran ParFam with each optimizer for seven different random seeds and different numbers of iterations. As we solely compare the influence of different optimizers in this experiment, we assume full knowledge of the perfect model parameters for each algorithm. Hence, we are only learning the parameters $\theta$ of one parametric family $(f_\theta)_{\theta \in \mathbb{R}^m}$ instead of iterating through multiple ones as in the experiments in Section 3.1. Therefore, we have to omit the problem Feynman-test-17 since the perfect model parameters result in a parametric family with too many parameters to be optimized in a reasonable time and, thus, wasting unreasonable resources. The results are presented in Figure 4. These show the superiority of basin-hopping and BFGS with multi-start compared to all the other algorithms. While basin-hopping and BFGS with multi-start perform similarly well, it is notable that basin-hopping is less sensitive to the training time (and hence the number of iterations). Therefore, we chose basin-hopping in the main paper, although using BFGS with multi-start would have led to similar results.

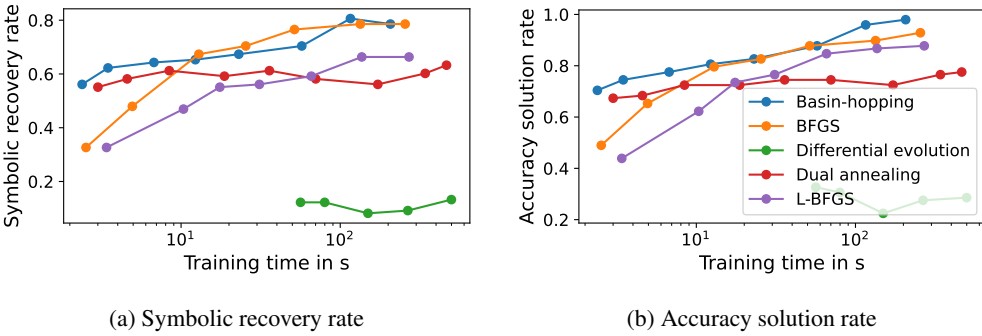

(a) Symbolic recovery rate          (b) Accuracy solution rate

Figure 4: Symbolic recovery and accuracy solution rate (percentage of data sets with $R^2 > 0.999$ for the test set) of ParFam with different optimizers on the subset of the Feynman problems displayed in Table 2.

## C   EXAMPLE FEYNMAN PROBLEMS

Table 2 shows a random subset of the Feynman data set. The complete Feynman data set can be downloaded from https://space.mit.edu/home/tegmark/aifeynman.html.

Table 2: Random subset of 15 equations of the Feynman problem set (Udrescu & Tegmark, 2020).

| Name | Formula |
| --- | --- |
| Feynman-III-4-33 | $y = \dfrac{h\omega}{2\pi \left( \exp\left( \frac{h\omega}{2\pi T k b} \right) - 1 \right)}$ |
| Feynman-III-8-54 | $y = \sin^2\left( \dfrac{2\pi E_n t}{h} \right)$ |
| Feynman-II-15-4 | $y = -Bmom \cos(\theta)$ |
| Feynman-II-24-17 | $y = \sqrt{-\dfrac{\pi^2}{d^2} + \dfrac{\omega^2}{c^2}}$ |
| Feynman-II-34-29b | $y = \dfrac{2\pi B J z g m o m}{h}$ |
| Feynman-I-12-5 | $y = E f q_2$ |
| Feynman-I-18-4 | $y = \dfrac{m_1 r_1 + m_2 r_2}{m_1 + m_2}$ |
| Feynman-I-38-12 | $y = \dfrac{\epsilon h^2}{\pi m q^2}$ |
| Feynman-I-39-22 | $y = \dfrac{T k b n}{V}$ |
| Feynman-I-40-1 | $y = n_0 \exp\left( -\dfrac{g m x}{T k b} \right)$ |
| Feynman-I-43-31 | $y = T k b m o b$ |
| Feynman-I-8-14 | $y = \sqrt{(-x_1 + x_2)^2 + (-y_1 + y_2)^2}$ |
| Feynman-I-9-18 | $y = \dfrac{G m_1 m_2}{(-x_1 + x_2)^2 + (-y_1 + y_2)^2 + (-z_1 + z_2)^2}$ |
| Feynman-test-17 | $y = \dfrac{m^2 \omega^2 x^2 \left( \frac{\alpha x}{y} + 1 \right) + p^2}{2m}$ |
| Feynman-test-18 | $y = \dfrac{3 \left( H_G^2 + \frac{c^2 k_f}{r^2} \right)}{8\pi G}$ |

# D  STROGATZ PROBLEMS

Table 3 shows the complete Strogatz data set. It can be downloaded from https://github.com/lacava/ode-strogatz.

Table 3: The Strogatz ODE problem set (La Cava et al., 2016).

| Name | Formula |
|---|---|
| Bacterial Respiration | $\dot{x} = -\frac{xy}{0.5x^2+1} - x + 20$ |
|  | $\dot{y} = -\frac{xy}{0.5x^2+1} + 10$ |
| Bar Magnets | $\dot{x} = -\sin(x) + 0.5\sin(x-y)$ |
|  | $\dot{y} = -\sin(y) - 0.5\sin(x-y)$ |
| Glider | $\dot{x} = -0.05x^2 - \sin(y)$ |
|  | $\dot{y} = x - \frac{\cos(y)}{x}$ |
| Lotka-Volterra interspecies dynamics | $\dot{x} = -x^2 - 2xy + 3x$ |
|  | $\dot{y} = -xy - y^2 + 2y$ |
| Predator Prey | $\dot{x} = x\left(-x - \frac{y}{x+1} + 4\right)$ |
|  | $\dot{y} = y\left(\frac{x}{x+1} - 0.075y\right)$ |
| Shear Flow | $\dot{x} = \cos(x)\cot(y)$ |
|  | $\dot{y} = \left(0.1\sin^2(y) + \cos^2(y)\right)\sin(x)$ |
| van der Pol oscillator | $\dot{x} = -\frac{10x^3}{3} + \frac{10x}{3} + 10y$ |
|  | $\dot{y} = -\frac{x}{10}$ |

# E  MODEL PARAMETER SEARCH

The success of ParFam depends strongly on a good choice of the model parameters: The set of base functions $g_1, ..., g_k$ and the degrees $d_i^1$ and $d_i^2$, $i \in \{1, \ldots, k+1\}$, of the numerator and denominator polynomials of $Q_1, ..., Q_{k+1}$, respectively. On the one hand, choosing the degrees very small or the set of base functions narrow might restrict the expressivity of ParFam too strongly and exclude the target function from its search space. On the other hand, choosing the degrees too high or a very broad set of base functions can yield a search space that is too high-dimensional to be efficiently handled by a global optimization method. This might prevent ParFam from identifying even very simple functions.

To balance this tradeoff, we allow ParFam to iterate through many different choices for the hyperparameters describing the model. The user specifies upper bounds on the degrees $d_i^1$ and $d_i^2$ of the polynomials and the set of base functions $g_1, \ldots, g_k$. ParFam then automatically traverses through different settings, starting from simple polynomials to rational functions to more complex structures involving the base functions and ascending degrees of the polynomials. The exact procedure is shown in Algorithm 1. Note that we refer to the rational functions $Q_1, ..., Q_k$, which will be the inputs to the base functions, as the 'input rationals' and, therefore, describe the degrees of their polynomials by 'DegInputNumerator' and 'DegInputDenominator'. Similarly, we denote the degrees of the polynomials of the output rational function $Q_{k+1}$ by 'DegOutputNumerator' and 'DegOutputDenominator'.

---

**Algorithm 1:** Traversal of the model parameters

---

**Input:** Maximal Degree Input Numerator $d^1_{\mathsf{max,in}}$,
    Maximal Degree Output Numerator $d^1_{\mathsf{max,out}}$,
    Maximal Degree Input Denominator $d^2_{\mathsf{max,in}}$,
    Maximal Degree Output Denominator $d^2_{\mathsf{max,out}}$,
    Maximal number of base functions $b_{\mathsf{max}}$
    Set of base functions $G_{\mathsf{max}} = \{g_1, \ldots, g_k\}$.
**Output:** List of model parameters $\mathcal{L}$ that define the models ParFam can iterate through.

---

1 Let $\mathcal{L} = \{\,\}$ be an empty list.
    // Start with a polynomial model:
2 $D_{\mathsf{p}}$ = {'DegInputNumerator': 0, 'DegOutputNumerator': $d^1_{\mathsf{max,out}}$, 'DegInputDenominator': 0,
    'DegOutputDenominator': 0, 'baseFunctions': []}
3 $\mathcal{L}$.append($D_0$)
    // Continue with purely rational models with different degrees:
4 **for** $d^2_{\mathsf{out}} = 1$ **to** $d^2_{\mathsf{max,out}}$ **do**
5 $\quad$ **for** $d^1_{\mathsf{out}} = 1$ **to** $d^1_{\mathsf{max,out}}$ **do**
6 $\quad\quad$ $D_{\mathsf{r}}$ = {'DegInputNumerator': 0, 'DegOutputNumerator': $d^1_{\mathsf{out}}$, 'DegInputDenominator':
    $\quad\quad$ 0, 'DegOutputDenominator': $d^2_{\mathsf{out}}$, 'baseFunctions': []}
7 $\quad\quad$ $\mathcal{L}$.append($D_{\mathsf{r}}$)
8 $\quad$ **end**
9 **end**
    // Include different combinations of base functions:
10 **for** $b = 1$ **to** $b_{\mathsf{max}}$ **do**
11 $\quad$ **for** $d^2_{\mathsf{out}} = 0$ **to** $d^2_{\mathsf{max,out}}$ **do**
12 $\quad\quad$ **for** $d^1_{\mathsf{out}} = 1$ **to** $d^1_{\mathsf{max,out}}$ **do**
13 $\quad\quad\quad$ **for** $d^2_{\mathsf{in}} = 0$ **to** $d^2_{\mathsf{max,in}}$ **do**
14 $\quad\quad\quad\quad$ **for** $d^1_{\mathsf{in}} = 1$ **to** $d^1_{\mathsf{max,in}}$ **do**
15 $\quad\quad\quad\quad\quad$ **for** $B$ **as a list with** $b$ **elements of** $G_{\mathsf{max}}$ **do**
    $\quad\quad\quad\quad\quad\quad$ // Note that base functions can be contained in
    $\quad\quad\quad\quad\quad\quad\quad$ $B$ multiple times.
16 $\quad\quad\quad\quad\quad\quad$ $D$ = {'DegInputNumerator': $d^1_{\mathsf{in}}$, 'DegOutputNumerator': $d^1_{\mathsf{out}}$,
    $\quad\quad\quad\quad\quad\quad$ 'DegInputDenominator': $d^2_{\mathsf{in}}$, 'DegOutputDenominator': $d^2_{\mathsf{out}}$,
    $\quad\quad\quad\quad\quad\quad$ 'baseFunctions': $B$}
17 $\quad\quad\quad\quad\quad\quad$ $\mathcal{L}$.append($D$)
18 $\quad\quad\quad\quad\quad$ **end**
19 $\quad\quad\quad\quad$ **end**
20 $\quad\quad\quad$ **end**
21 $\quad\quad$ **end**
22 $\quad$ **end**
23 **end**
24 **return** $\mathcal{L}$

---

This strategy is comparable to the one proposed by Bartlett et al. (2023), called "Exhaustive Symbolic Regression". There, they iterate through a list of parameterized functions and use BFGS to identify the parameters. To create the list of parametrized functions, they construct every possible function using a given set of base operations and a predefined complexity. Notably, this results in more than 100,000 functions to evaluate for one-dimensional data, with the same set of base functions as we do, but without $\cos$. Our algorithm, however, only needs to search for the parameters of around 500 functions since it covers many at the same time by employing the global optimization strategy.

Due to this high complexity, Bartlett et al. (2023) state that they merely concentrate on one-dimensional problems and, thus, could benchmark their algorithm only on Feynman-I-6-2a ($y = \exp(\theta^2/2)/\sqrt{2pi}$), the only one-dimensional problem from the Feynman data set (Udrescu &

Tegmark, 2020). This example shows the benefit of employing global search in the parameter space: While ParFam needs five minutes of CPU time to compute the correct function, Bartlett et al. (2023) need 33 hours (150 hours, if the set of possible functions is not pre-generated).

## F  HYPERPARAMETER SETTINGS SRBENCH GROUND-TRUTH PROBLEMS

The hyperparamater settings for the SRBench ground-truth problems are summarized in Table 4.

Table 4: The model and optimization parameters for the SRBench ground-truth problems

| | | |
|---|---|---|
| **Model parameters** | Maximal Degree Input Numerator | 2 |
| | Maximal Degree Input Denominator | 2 |
| | Maximal Degree Output Numerator | 4 |
| | Maximal Degree Input Denominator | 3 |
| | Base functions | $\sqrt{}$, cos, exp |
| | Maximal potence of any variable (i.e., $x_1^4$ is excluded but $x_1^3 x_2$ is allowed) | 3 |
| **Optimization parameters** | Global optimizer | Basin-hopping |
| | Local optimizer | BFGS |
| | Maximal number of iterations global optimizer | 10 |
| | Maximal data set length | 1000 |
| | Regularization parameter $\lambda$ | 0.001 |
| | Maximal runtime | 8 CPU hours |
| | Maximal number of evaluations | 1,000,000 |

## G  ADDITIONAL PLOTS FOR THE SRBENCH GROUND-TRUTH RESULTS

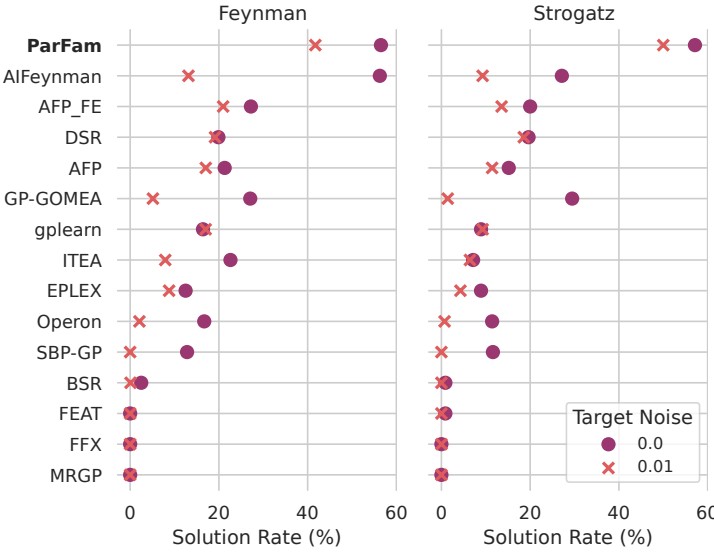

Figure 5: Symbolic recovery rate on both SRBench ground-truth data sets separated.

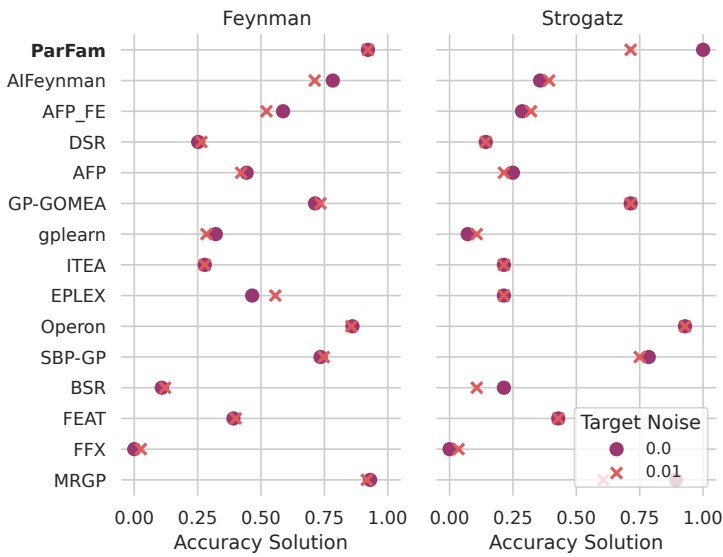

Figure 6: Accuracy solution rate (percentage of data sets with $R^2 > 0.999$ for the test set) on the SRBench ground-truth problems separately.

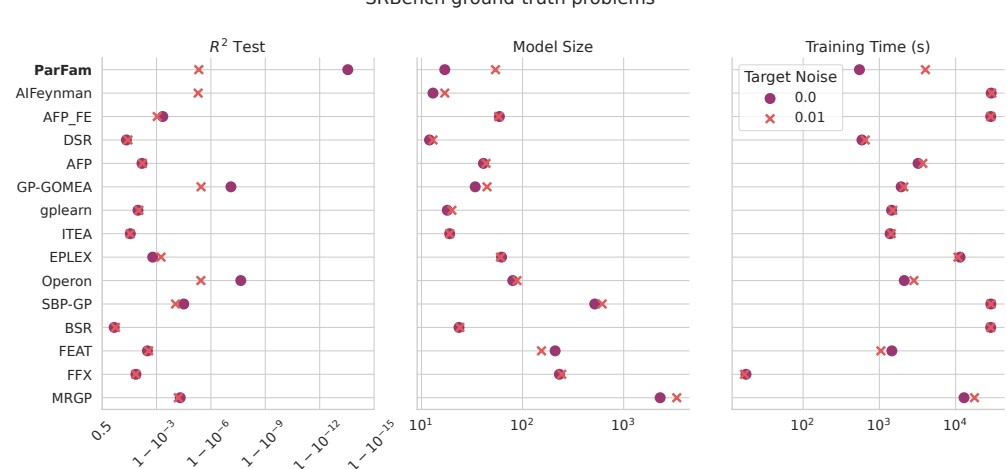

Figure 7: Results on the SRBench ground-truth problems. Points indicate the median test set performance on all problems. The $R^2$ Test for AIFeynman is missing on the plot since SRBench used a higher precision data type, such that AIFeynman achieved a median $R^2$ greater than $1 - 10^{-16}$.

## H  SENSITIVITY ANALYSIS FOR $\lambda$

In Table H, we present the results for ParFam on the ground-truth SRBench data sets for different values of $\lambda$. Note, that this has been done afterwards as a sensitivity analysis and not to choose $\lambda$. Our selection of $\lambda = 0.001$ was based on theoretical considerations and prior observations on toy examples while debugging ParFam.

## I  COMPARING PARFAM TO EQL ON SRBENCH

As described in the introduction, EQL (Martius & Lampert, 2017; Sahoo et al., 2018) is the closest method to ParFam, since both make use of non-linear parametric models to translate SR to a continuous optimization problem. Because of this similarity, we believe that it is important to also

Table 5: Results of ParFam on the ground-truth SRBench data sets for different values of $\lambda$.

| $\lambda$ | Accuracy solution rate | Symbolic recovery rate | Complexity |
|---|---|---|---|
| 0.0001 | 94.7% | 50% | 227 |
| 0.001 | 91.7% | 55.6% | 131 |
| 0.01 | 94.7% | 52.2% | 243 |

Table 6: Results of ParFam and EQL (Martius & Lampert, 2017; Sahoo et al., 2018) on the 96 SRBench ground-truth equations, which do not include the square root, logarithm, and exponential.

| | Accuracy solution rate | Symbolic recovery rate |
|---|---|---|
| ParFam | 93.8% | 69.8% |
| EQL | 75% | 16.7% |

show numerical comparisons between these two. However, EQL is not able to express the square root, logarithm, and exponential, which is why we created a reduced version of the ground-truth SRBench, which omits all equations using any of these base functions. In total, this covers 96 formulas. The results on these can be seen in Table 6.

To ensure a fair comparison for EQL, we first tried to run it using the default learning parameters and model parameter search recommended by the authors. However, since EQL will then quickly use up the computing budget given by SRBench (8 hours of CPU time) we tested EQL for multiple different hyperparameters on the first 20 problems from SRBench. We then chose the hyperparameters for which this worked the best and reran the whole benchmark. This, together with the initial run using the recommended parameters, gives two formulas per equation. In the results shown in Table 6 we chose the formula with the better $R^2$ on the validation data set. Note, that we did not make use of the information, that the square root, logarithm, and exponential are never parts of the formulas when running ParFam, i.e., we included these base functions in the dictionary.

## J  NGUYEN BENCHMARK

To compare ParFam with SPL (Sun et al., 2022) and NGGP (Mundhenk et al., 2021), which are the current state-of-the-art on some SR benchmarks (like Nguyen (Uy et al., 2011)), but no results of them on SRBench were reported, we evaluate ParFam on Nguyen. Interestingly, we observed that the original domain on which the data was sampled is not big enough to specify the functions, as ParFam was able to find simple and near indistinguishable approximations to the data that are not the target formula. For example, it found $0.569x^2 - 0.742\sin(1.241x^2 - 2.059) - 1.655$ instead of $\sin(x^2)\cos(x) - 1$, since both are almost identical on the domain $[-1, 1]$. For this reason, we extended the data domain for some of the problems. The results for the Nguyen data set can be seen in Table 7. We used the hyperparameters shown in Table 8. Following Sun et al. (2022), from whom we take the results of the competitors, we use $\sin$ and $\exp$ as the standard basis functions for ParFam and add $\sqrt{}$ and $\log$ for the problems 7, 8, 11, and $8^c$. Note that formula Nguyen-11 can not be expressed by ParFam and hence the symbolic recovery rate is 0.

Table 7: Results on the Nguyen benchmarks. The results for ParFam are averaged over 6 independent runs. The results from SPL (Sun et al., 2022), NGGP (Mundhenk et al., 2021), and GP (a genetic programming based SR algorithm) are taken from Sun et al. (2022).

| Benchmark | Expression | ParFam | SPL | NGGP | GP |
|---|---|---|---|---|---|
| Nguyen-1 | $x^3 + x^2 + x$ | 100% | 100% | 100% | 99% |
| Nguyen-2 | $x^4 + x^3 + x^2 + x$ | 100% | 100% | 100% | 90% |
| Nguyen-3 | $x^5 + x^4 + x^3 + x^2 + x$ | 100% | 100% | 100% | 34% |
| Nguyen-4 | $x^6 + x^5 + x^4 + x^3 + x^2 + x$ | 100% | 99% | 100% | 54% |
| Nguyen-5 | $\sin\left(x^2\right)\cos\left(x\right) - 1$ | 83% | 95% | 80% | 12% |
| Nguyen-6 | $\sin\left(x\right) + \sin\left(x^2 + x\right)$ | 83% | 100% | 100% | 11% |
| Nguyen-7 | $\log\left(x + 1\right) + \log\left(x^2 + 1\right)$ | 100% | 100% | 100% | 17% |
| Nguyen-8 | $\sqrt{x}$ | 100% | 100% | 100% | 100% |
| Nguyen-9 | $\sin\left(x_0\right) + \sin\left(x_1^2\right)$ | 100% | 100% | 100% | 76% |
| Nguyen-10 | $2\sin\left(x_0\right)\cos\left(x_1\right)$ | 100% | 100% | 100% | 86% |
| Nguyen-11 | $x^y$ | 0% | 100% | 100% | 13% |
| Nguyen-12 | $x_0^4 - x_0^3 + 0.5x_1^2 - x_1$ | 100% | 28% | 4% | 0% |
| Nguyen-1$^c$ | $3.39x^3 + 2.12x^2 + 1.78x$ | 100% | 100% | 100% | 0% |
| Nguyen-2$^c$ | $0.48x^4 + 3.39x^3 + 2.12x^2 + 1.78$ | 100% | 94% | 100% | 0% |
| Nguyen-5$^c$ | $\sin\left(x^2\right)\cos\left(x\right) - 0.75$ | 83% | 95% | 98% | 1% |
| Nguyen-8$^c$ | $\sqrt{1.23x}$ | 100% | 100% | 100% | 56% |
| Nguyen-9$^c$ | $\sin\left(1.5x_0\right) + \sin\left(0.5x_1^2\right)$ | 100% | 96% | 90% | 0% |
| Average | | 91.2% | 94.5% | 92.4% | 38.2% |

Table 8: The model and optimization parameters for the Nguyen benchmark.

| | | |
|---|---|---|
| **Model parameters** | Maximal Degree Input Numerator | 2 |
| | Maximal Degree Input Denominator | 0 |
| | Maximal Degree Output Numerator | 6 |
| | Maximal Degree Input Denominator | 0 |
| | Base functions | cos, exp ($\sqrt{\phantom{x}}$, log) |
| | Maximal potence of any variable | 6 |
| **Optimization parameters** | Global optimizer | Basin-hopping |
| | Local optimizer | BFGS |
| | Maximal number of iterations global optimizer | 30 |
| | Regularization parameter $\lambda$ | 0.1 |

## K   SYNTHETIC DATA SET

The synthetic data set for the training and evaluation of DL-ParFam in Section 3.2 is sampled in the following way. We first create a parametric family $(f_\theta)_{\theta \in \mathbb{R}^m}$ for fixed model hyperparameters, i.e., for a specific choice of base functions and maximal degrees. For the experiments, we choose the degree of all denominator polynomials to be 0, i.e., $Q_1, ..., Q_{k+1}$ are simple polynomials, and their maximal degree will be set to 2. As the base functions, we use sin and $\sqrt{}$.

Now, we aim to sample sparse parameters $\theta \in \mathbb{R}^m$ to obtain interpretable functions. To achieve this, we randomly choose the number of non-zero coefficients of $Q_{k+1}$ to be 1 or 2. Next, we choose those functions among $Q_1, ..., Q_k$ which are used by $Q_{k+1}$ and randomly select the non-zero coefficients of these. If only one coefficient is chosen, we ensure that it does not correspond to the constant term. For each non-zero coefficient chosen that way, we sample a parameter $\theta_i \sim \mathcal{N}(0, 9)$.

This way we sample a set of parameters $(\theta^j)_{j=1,...,K}$ describing (interpretable) functions $f_{\theta^j}$. These functions are evaluated on the fixed grid $x = -10, -9.9, -9.8, \ldots, 9.9$ to create the data sets $((x_i)_{i=1,...,m}, (y_i^j = f_{\theta^j}(x_i))_{i=1,...,m})_{j=1,...,K}$. Some examples for $f_{\theta^j}$ are presented in Table 9.

Table 9: Example formulas from the synthetic data set used to train and evaluate DL-ParFam in Section 3.2. All coefficients are rounded to three decimal places.

| | Formula |
|---|---|
| 1. | $y = 1.357 \sin\left(4.072x^2 + 1.044\right)$ |
| 2. | $y = -2.909x \sin\left(8.746x^2 - 1.637x + 0.72\right) + 2.591$ |
| 3. | $y = 4.131x \sin\left(1.933x^2 - 0.549x + 3.205\right) - 2.847 \sin\left(1.933x^2 - 0.549x + 3.205\right)\sqrt{|1.344x + 1.678|}$ |
| 4. | $y = 0.771x\sqrt{|3.293x^2 + 0.878x + 1.837|} + 3.64 \sin\left(0.824x^2 - 8.78x + 1.936\right)$ |
| 5. | $y = -1.375x$ |
| 6. | $y = -6.339x \sin\left(6.961x\right) + 2.891$ |
| 7. | $y = 0.944x \sin\left(2.042x^2 + 5.451x + 3.97\right) + 0.548x$ |
| 8. | $y = -3.907x \sin\left(5.384x^2\right) + 2.681 \sin\left(5.384x^2\right)\sqrt{|0.276x^2 + 2.406x - 1.149|}$ |
| 9. | $y = 4.276x + 2.025 \sin\left(3.93x\right)$ |
| 10. | $y = -3.007x\sqrt{|x^2|} - 2.751 \sin\left(0.231x^2 - 1.3\right)$ |
| 11. | $y = -2.189x^2 - 2.828 \sin\left(0.188x^2 + 0.63\right)$ |
| 12. | $y = -0.317 \sin\left(-2.351x^2 + 1.448x + 2.344\right) - 6.667$ |
| 13. | $y = 2.154x + 3.064 \sin\left(4.773x^2\right)\sqrt{|4.491x - 0.423|}$ |
| 14. | $y = -0.772x^2 + 0.333x \sin\left(2.938x^2 + 2.245x\right)$ |
| 15. | $y = -6.929 \sin\left(2.451x^2 + 2.37x + 4.415\right)\sqrt{|2.486x - 1.428|} - 1.873 \sin\left(2.451x^2 + 2.37x + 4.415\right)$ |
| 16. | $y = -0.528x^2 - 0.601 \sin\left(3.249x^2\right)$ |
| 17. | $y = -3.77 \sin\left(0.577x\right)\sqrt{|0.981x - 2.192|}$ |
| 18. | $y = 1.076 \sin\left(0.189x\right)$ |
| 19. | $y = -2.046x$ |
| 20. | $y = 1.608 \sin\left(5.982x - 2.644\right)\sqrt{|x^2|}$ |

## L   HYPERPARAMETER SETTINGS SYNTHETIC BENCHMARK

The hyperparamater settings for the experiments on the synthetic benchmark in Section 3.2 are summarized in Table 10.

Table 10: The model and optimization parameters for the benchmark with the synthetic data sets.

| | | |
|---|---|---|
| **Data set parameters** | Training set size | 2,000,000 |
| | Validation set size | 10,000 |
| | Test set size (for the NN) | 10,000 |
| | Test set size (for DL-ParFam vs ParFam) | 100 |
| | Maximal number of non-zero coefficients of $Q_{k+1}$ | 2 |
| **Model parameters** **(data creation and training)** | Maximal Degree Input Numerator | 2 |
| | Degree Input Denominator | 0 |
| | Degree Output Numerator | 2 |
| | Degree Input Denominator | 0 |
| | Base functions | $\sqrt{\ }$, sin |
| | Maximal potence of any variable | 2 |
| **Optimization parameters** **(Neural network pre-training)** | Optimizer | ADAM |
| | Loss | BCE |
| | Number epochs | 20,000 |
| | Number batches | 500 |
| | Number hidden layers | 1 |
| | Number hidden neurons per layer | 200 |
| | Learning rate | 0.0001 |
| **Optimization parameters** **(ParFam and DL-ParFam)** | Global optimizer | Basin-hopping |
| | Local optimizer | BFGS |
| | Maximal number of iterations global optimizer | 10, 20, 50, 100 |
| | Regularization parameter $\lambda$ | 0.001 |
| | Maximal runtime | no limit |
| | Maximal number of evaluations | no limit |

