# OpenReview forum: "ParFam - Symbolic Regression Based on Continuous Global Optimization"
_ICLR.cc/2024/Conference — Submitted to ICLR 2024_

### Official Review · Reviewer_1Ff3 · 2023-10-18

**Soundness:** 1 poor
**Presentation:** 3 good
**Contribution:** 2 fair
**Rating:** 5
**Confidence:** 4

**Summary:**

This paper introduces a simple parametric method for symbolic regression, as well as a deep learning-based extension where the neural network is designed to constrain the set of learnable parameters.

**Strengths:**

- Results on SRbench black-box problems: the Parfam method seems competitive
- Clarity: the paper is well written and easy to follow

**Weaknesses:**

- Lack of novelty: the parametric approach is not particularly novel (it is used in existing methods such as EQLearner and FFX). The main trick enabling the competitive performance seems to rely a lot on manual crafting of the heuristics (Appendix A) and the extensive model parameter search (Appendix E). As for DL-parfam, it is not sufficiently validated, as detailed below.
- Experimental validation: as acknowledged by the authors, the DL-parfam method is mainly in prototype stage right now. Are results of DL-parfam on Feynman problems not reported because they were not as good as the ones on synthetic data or because the authors did not have the time to test? In the first case, the authors should at least explain why the results aren’t good (what is missing in the current state). In the second, it gives the paper an unfinished impression. In both cases, this section appears as a dealbreaker for a prestigious venue — results should be complete, otherwise the paper appears rushed.

**Questions:**

"Even though modern approaches are able to handle flexible data sets in high dimensions (Biggio et al., 2021; Kamienny et al., 2022), they fail to incorporate invariances in the function space, e.g., x + y and y + x are seen as different functions, as pointed out by Holt et al. (2023)"

I tend to disagree with the idea that this is the main limitation of modern approaches. In fact, modern methods easily learns these invariances, which can be seen by the fact that beam search typically reveals equivalent expressions.

---

> ### Author Response · Authors · 2023-11-20
> **Official Comment by Authors (1/2)**
>
> We thank Reviewer 1Ff3 for their time and thoughtful comments.
>
> __Lack of novelty: the parametric approach is not particularly novel (it is used in existing methods such as EQLearner and FFX). The main trick enabling the competitive performance seems to rely a lot on manual crafting of the heuristics (Appendix A) and the extensive model parameter search (Appendix E).__
>
> Thank you for raising this important topic. In our general response, we addressed the similarity to EQL and added a more in-depth discussion at the end of the introduction.
>
> FFX also follows the idea of translating SR to a continuous optimization problem using parametric functions. However, the main difference between FFX and ParFam is that FFX is limited to a parametric function which is linear in its parameters, to be able to use efficient regression techniques. This, however, limits the search space strongly, since they cannot model any coefficients inside of the base functions, i.e., functions like $\sin(ax)$ are impossible. We decided to allow a more expressive parametric family for the cost of having a more complicated optimization problem. This difference in expressivity can also be seen in the results on the SRBench data sets. Since we agree that it is important to place ParFam in the field of SR and also show its differences with similar methods we added the following paragraph in the introduction regarding parametric SR models in general and FFX and SINDy specifically:
>
> > Most SR algorithms approach the problem by first searching for the analytic form of $f$ and then optimizing the resulting coefficients. In contrast, only a few algorithms follow the same idea as ParFam, to merge these steps into one by spanning the search space using an expressive parametric model and searching for sparse coefficients that simultaneously yield the analytical function and its coefficients. FFX (McConaghy, 2011) and SINDy (Brunton et al., 2016) utilize a model to span the search space which is linear in its parameters, to be able to apply efficient methods from linear regression to compute the coefficients. To increase the search space, they construct a large set of features by applying the base functions to the input variables. While these linear approaches enable fast processing in high dimensions, they are unable to model non-linear parameters within the base functions, restricting the search space to a predefined set of features.
>
> Regarding the question of what makes ParFam competitive: We do not believe that it is due to the manual crafting of heuristics and an extensive model parameter search. FFX is strongly limited by its expressivity since it only allows functions linear in the parameters, so manual crafting of heuristics and an extensive model parameter search would not be sufficient to reach competitive results on a complicated benchmark like SRBench. For our experiments with EQL on SRBench, we followed the general setting of Sahoo et al. [1] and performed also a model parameter search. However, due to the complicated optimization process, as discussed in our general response, the training time for each hyperparameter set is too high to cover multiple ones on SRBench, with their specified hardware settings (8h CPU times). We include experiments on a comparison between EQL and ParFam as soon as they are finished, see the request by Reviewer 5PDr. Furthermore, notice that the heuristics in Appendix A are mainly to avoid numerical issues, for instance, to avoid division by near zero terms and to avoid negative inputs to the square root.
>
> __As for DL-parfam, it is not sufficiently validated__
>
> Thank you for raising this important concern. We aimed to explain this in the general response and added more clarification in the paper. So, to answer the question of why the results on SRBench are currently not reported is because the method is at its current stage not able to handle the data sets given as inputs. This is due to the variability of the input dimension and the number of data points. We are thankful to Reviewer 1Ff3 for pointing this out and added a clarification of the prototype state of DL-ParFam at multiple points (see the main reply) and an explanation of why the SRBench data sets are out of reach for our prototype of DL-ParFam currently in Section 3.2:
>
> > Due to the prototype status of DL-ParFam, the ability to evaluate it on complex data sets, such as the Feynman dataset, is limited as the data to be processed is not sampled on the same grid. Therefore, we use synthetic data sets.

---

> > ### Author Response · Authors · 2023-11-20
> > **Official Comment by Authors (2/2)**
> >
> > __"Even though modern approaches are able to handle flexible data sets in high dimensions (Biggio et al., 2021; Kamienny et al., 2022), they fail to incorporate invariances in the function space, e.g., x + y and y + x are seen as different functions, as pointed out by Holt et al. (2023)"
> > I tend to disagree with the idea that this is the main limitation of modern approaches. In fact, modern methods easily learns these invariances, which can be seen by the fact that beam search typically reveals equivalent expressions.__
> >
> > This is a very interesting point, thank you for catching up on this. We think the idea of Holt et al. (2023) was that it is true that these methods can learn the invariances in the end, i.e., predict multiple equivalent functions via beam search, however, at training time the symbolic functions are never evaluated, and just compared symbolically, making the training process more complicated. This means if there are two equivalent functions in the training data the neural network can never know which one is correct. Therefore, even a correct prediction with the "wrong" symbolic formula will yield a loss that will cause the parameters to change according to that equation even though it predicted the perfect formula here. This happens because the beam search and the evaluation of the formula afterward are never used. To clarify this we added the following
> >
> > > Even though modern approaches are able to handle flexible data sets in high dimensions (Biggio et al., 2021; Kamienny et al., 2022), they fail to incorporate invariances in the function space *during training*, e.g., $x+y$ and $y+x$ are seen as different functions, as pointed out by Holt et al. (2023)*, which possibly complicates the training*.
> >
> > We again want to thank Reviewer 1Ff3 for their time and hope we could address all their concerns.
> >
> > [1] Subham Sahoo, Christoph Lampert, and Georg Martius. Learning equations for extrapolation and control. In Proceedings of the 35th International Conference on Machine Learning, volume 80 of Proceedings of Machine Learning Research, pp. 4439–4447. PMLR, 2018. URL http://proceedings.mlr.press/v80/sahoo18a.html.

---

> > > ### Comment · Reviewer_1Ff3 · 2023-11-21
> > > **Response to rebuttal**
> > >
> > > Thanks to the authors for the response!
> > > I will maintain my score at the same level, because I believe that the contribution of Parfam is not sufficient to lead to publication alone, and the DL-Parfam method is still prototyped. But should the paper be rejected, I strongly encourage the authors to resubmit once DL-Parfam is improved!

---

### Official Review · Reviewer_8mnU · 2023-10-31

**Soundness:** 2 fair
**Presentation:** 2 fair
**Contribution:** 2 fair
**Rating:** 6
**Confidence:** 2

**Summary:**

This paper considers the task of symbolic regression that learns to discover the underlying expression from data. The authors make an important observation that the current expression is just a small fraction of the whole possible expression, so searching in this small family would be much easier than searching in the whole space. The author justifies the success of the proposed on several datasets and many baselines.

**Strengths:**

- The idea is clearly written and the observation for the current symbolic regression dataset is interesting.
- The experiment result is strong against a lot of baselines.

**Weaknesses:**

- Figure 1 as well as the description of Equations 3 and 4 are very hard to understand. It is unclear how the observation in Equation 1 is actually implemented into an algorithm.
- A deep understanding of the proposed family of symbolic expressions is needed. Since the observation is so strong, it eliminates a lot of "impossible" expressions and reduces the search space greatly. I hope the author could give some analysis on how much the reduction of the search space from all the possible expressions (of maximum length < 30) to the family of expressions described in Equation 1.
- One important baseline is missing: Symbolic physics learner: Discovering governing equations via Monte Carlo tree search.
- The basin-hopping algorithm is used to solve non-convex optimization problems. Is the structure of the symbolic family required to solve non-convex optimization instead of convex optimization? This is not justified. Also `scipy.optimize` has already offered the API for BFGS, basin-hopping, SHGO, Direct, dual annealing, and Differential evolution. The whole process of using these fancy optimizers is just changing these APIs in one line.

Here is the link: https://docs.scipy.org/doc/scipy/reference/optimize.html

**Questions:**

1. A detailed description of the pipeline in Figure 1 is needed.
2. Theoretically analysis of the observation of Equation 1 on the reduction of space of symbolic regression is needed.

---

> ### Author Response · Authors · 2023-11-20
>
> We thank Reviewer 8mnU for their time and thoughtful comments on our paper and a chance to address their concerns which we do in the following:
>
> __Figure 1 as well as the description of Equations 3 and 4 are very hard to understand. A detailed description of the pipeline in Figure 1 is needed__
>
> Thank you for pointing this out. We added a more detailed description to Figure 1 in our paper.
>
> Equation 3 is the definition of the loss function of the neural network employed for DL-ParFam as the sum of the binary-cross entropy loss and Equation 4 is the standard definition of the [binary-cross entropy loss](https://pytorch.org/docs/stable/generated/torch.nn.BCELoss.html). Please let us know if there is still a problem regarding the clarity of these.
>
> __It is unclear how the observation in Equation 1 is actually implemented into an algorithm.__
>
> Equation 1 defines the structure of the parametric function $f_\theta$ and the goal of ParFam is to find sparse parameters $\theta$ such that $f_\theta(x_i)\approx y_i$. This is done by minimizing the loss function defined in Equation 2.
>
> __I hope the author could give some analysis on how much the reduction of the search space from all the possible expressions (of maximum length < 30) to the family of expressions described in Equation 1. Theoretically analysis of the observation of Equation 1 on the reduction of space of symbolic regression is needed.__
>
> Thank you for raising this very interesting question. ParFam can represent any function that can be represented by an expression such that each path from the root to the leaf contains at most one unary base function (e.g., $\sin$, $\cos$, $\exp$, $\sqrt{}$, etc.). We are working on determining the exact ratio of functions covered by ParFam currently and will provide further updates on this the next days since we agree that this is an insightful theoretical analysis. However, we argue that this should not be taken solely as a measure of expressivity since this would give the same importance to the formula $\sin(\sin(...(\sin(x_1))))$ as to $x_1+x_2$ where the latter one is more common in the real world and more interpretable. The focus of ParFam was to focus on the functions showing these properties: interpretable and common in the real world, see also our general response. Therefore, keeping this measure small restricts ParFam on the one hand, however, also strengthens it since the search space is strongly reduced, biased towards interpretable and common formulas.
>
> Apart from the question how many formulas can be exactly represented by ParFam, one might also be interested in its approximative capabilities. ParFam can approximate any continuous function on a compact set which follows directly from the [Stone-Weierstrass Theorem](https://en.wikipedia.org/wiki/Stone%E2%80%93Weierstrass_theorem). I hope we could address your concern reasonably well.
>
> __One important baseline is missing: Symbolic physics learner: Discovering governing equations via Monte Carlo tree search.__
>
> Thank you for pointing us to this very interesting work. Unfortunately, they do neither report their performance on SRBench and nor make their code available such that we can perform these experiments on our own. To be able to compare ParFam to SPL we have to use the benchmarks used in their paper. The only standard benchmark they used is the Nguyen benchmark with and without constants, for which we will report our results as soon as they are finished in our supplementary material.
>
> __The basin-hopping algorithm is used to solve non-convex optimization problems. Is the structure of the symbolic family required to solve non-convex optimization instead of convex optimization? This is not justified.__
>
> Since some of the unary-base functions (e.g., $\sin, \cos, \sqrt{}$) are non-convex and rational functions are not necessarily convex in their parameters, the loss function of ParFam can be expected to be non-convex for most model parameter choices and target functions. E.g., approximating $\cos(ax)$ with $f_\theta(x)=\theta_1x+cos(\theta_2x)+\theta_3$ results in a highly non-convex loss function. This can also be seen by the difference in performance between the optimizers in Section B, since a convex problem would have been solved by all of those using gradient information.
>
> __Also `scipy.optimize` has already offered the API for BFGS, basin-hopping, SHGO, Direct, dual annealing, and Differential evolution. The whole process of using these fancy optimizers is just changing these APIs in one line.__
>
> Yes, the goal of Appendix B was to give a comparison of different global optimizers, to show that the choice of basin hopping is optimal but not necessary for ParFam to work well and that simple heuristics like BFGS with multi-start work reasonably well themselves.
>
> We thank Reviewer 8mnU again and hope we could clarify their concerns!

---

> > ### Author Response · Authors · 2023-11-22
> > **Theoretical analysis of the search space**
> >
> > We thank Reviewer 8mnU again for pointing us to the interesting theoretical analysis of the search space. In the following, we want to provide the ideas and results:
> >
> > Assume that we consider expressions that can be represented as an expression tree using $n$ leaves (these are the variables and parameters), $k$ nodes with one child (these are the unary functions), and $b$ nodes with two children (these are binary operation like +, -, *, and /).
> >
> > For $i=0,1$: Denote by $B(l,i)$ the number of expression trees with $l$ nodes such that the path from the root to the leaf with the most unary functions has exactly $i$ unary functions.
> >
> > For $i=2$: Denote by $B(l,2)=$ the number of expression trees with $l$ nodes such that the path from the root to the leaf with the most unary functions has at least $2$ unary functions
> >
> > It is easy to see that $B(1,0)=n$ and $B(1,1)=B(1,2)=0$, i.e., all expression trees with 1 node do not have a path with 1 or more unary functions. This is trivial, since all expression trees only consist out of 1 leaf (from which there are $n$ many).
> >
> > For $l>1$, one might utilize the following formula to compute the values inductively:
> >
> > For $i=0$: $B(l,0)=b\sum_{l_1+l_2=l-1}B(l_1,0)B(l_2,0)$
> >
> > For $i=1$: $B(l,1)=b\sum_{i_1,i_2\leq 1; \max\{i_1,i_2\}=1; l_1+l_2=l-1}B(l_1,i_1)B(l_2,i_2)+kB(l-1,i-1)$
> >
> > For $i=2$: $B(l,2)=b\sum_{i_1,i_2\leq 2; \max\{i_1,i_2\}\geq 2; l_1+l_2=l-1}B(l_1,i_1)B(l_2,i_2)+k(B(l-1,1)+B(l-1,2))$
> >
> > The explanation for these formulas is the following:
> > - The first term with the sum in each formula, is the number of expression trees with $l$ and $i$ as before such that a binary operator is the root.
> > - The second term (if existent) is the number of expression trees with $l$ and $i$ as before such that a unary function is the root. This is the reason why it is only added for $i>0$.
> >
> > Computing general, non-recursive formulas, however, remains highly complex or even impossible. Instead one can harness computers to compute the $B(l,i)$ for high $l$. To get an idea how these look like we will show the values for $l=30$, as requested by Reviewer 8mnU:
> >
> > $B(30,0)=0$
> >
> > $B(30,1)=4 b^{10} k n^{11} \\cdot \\left(19389690 b^{4} n^{4} + 197732564 b^{3} k^{2} n^{3} + 208824567 b^{2} k^{4} n^{2} + 29057028 b k^{6} n + 359645 k^{8}\\right)$
> >
> > $B(30,2)=k^{3} n (1923626344 b^{13} n^{13} + 23867166792 b^{12} k^{2} n^{12} + 91635784968 b^{11} k^{4} n^{11} + 168210585400 b^{10} k^{6} n^{10} + 168212023980 b^{9} k^{8} n^{9} + 97045398450 b^{8} k^{10} n^{8} +$
> >
> > $33272708040 b^{7} k^{12} n^{7} + 6850263420 b^{6} k^{14} n^{6} + 841260420 b^{5} k^{16} n^{5} + 60090030 b^{4} k^{18} n^{4} + 2375100 b^{3} k^{20} n^{3} + 47502 b^{2} k^{22} n^{2} + 406 b k^{24} n + k^{26})$
> >
> > Since these are hard to interpret, we want to compute the explicit portion of the search space covered by ParFam. For this we set $C(l)=\sum_{l_1=1,...,l}(B(l_1,0) + B(l_1,1))$ as the number of expression trees with at most $l$ nodes ParFam can express and $D(l)=\sum_{l_1=1,...,l}(B(l_1,0) + B(l_1,1)+B(l_1,2)$ as the number of all expression trees with at most $l$ nodes. As in our experiments on SRBench we choose $b=4$ and $k=3$. In the following one can see the values for $C(l) / D(l)$ for different values of $l$ and $n$.
> >
> > |$n$ \ $l$| 3  | 6  | 9  | 12  | 15  | 18  | 21  |  24 | 27  | 30  |
> > |---|---|---|---|---|---|---|---|---|---|---|
> > |1| 47.059% | 26.763%  | 9.464%  | 3.873%  |  1.556% |  0.641% | 0.261%  |  0.108% | 0.044%  |  0.018% |
> > |5|  72.727% | 66.438%  | 34.271%  | 27.556%  | 16.943%  |  11.929% |  7.829% | 5.351%  | 3.575%  | 2.427%  |
> > |10| 83.019%  | 80.038%  | 47.999%  | 45.837%  |  30.388% | 25.583%  | 18.476%  |  14.664% | 10.972%  | 8.533%  |
> >
> >
> > Interestingly, for a fixed $n$, $C(l) / D(l)$ converges to an exponential for increasing $l$. E.g., for $n=1$, $\frac{C(l) / D(l)}{C(l-1) / D(l-1)}$ converges to ~0.744, for $n=5$ it converges to ~0.88, and for $n=10$ to 0.91.
> >
> > We hope that this helps to foster a theoretical understanding of the restriction of the search space and, therefore, shed light on the tradeoff between high expressivity and efficient computing.
> >
> > We are thankful for your comment and welcome the opportunity to deepen these theoretical considerations in further research.

---

### Official Review · Reviewer_LMH9 · 2023-10-31

**Soundness:** 2 fair
**Presentation:** 2 fair
**Contribution:** 2 fair
**Rating:** 5
**Confidence:** 3

**Summary:**

This work proposes ParFam, a simple regression method with a fixed and predefined structure, to tackle the symbolic regression problem. In ParFam, the function expression structure is directly specified by the user in advance, and then the coefficients are learned with a sparsity regularization from the observed data. In this way, the original symbolic regression problem can be reduced to a continuous optimization problem with respect to the coefficients.

Based on the ground-truth problems from SRBench and the knowledge from the Cambridge Handbook of Physics Formulation, this work proposes a reasonable parametric expression structure to represent the physical formulas. Then, it uses a global continuous optimization method (basin-hopping algorithm) to find the optimal coefficients of the predefined expression. Experimental results show that ParFam can achieve promising performance on the symbolic regression problem for physics formulas.

**Strengths:**

Symbolic regression (SR) is an important but difficult problem that can be found in various domains. The proposed ParFam method can achieve promising performance on SR for physics formulas in a straightforward way.

**Weaknesses:**

Although I enjoy reading this paper and appreciate the explicit discussion on the limitations, I have some major concerns about ParFam.

**1. Is It still Symbolic Regression?**

To my understanding, symbolic regression is a learning-based approach to find the mathematical expression of a function from the observed data, which includes two important components:

- Learn the analytical function structure;

- Optimize the coefficients (parameters) of the structure;

The former is unique for symbolic regression, which distinguishes it from the other regression problems. Symbolic regression is difficult and is currently shown to be HP-hard with formal proof [1]. I think this is the reason why an efficient (approximate) SR algorithm will be "usually quite complicated" as described in this work.

In ParFam, however, the analytical structure learning step is totally bypassed with a predefined function structure. The original problem is hence reduced to sparse regression with a fixed structure, and the only goal is to find the optimal coefficients. Is it still symbolic regression?

**2. Strong Prior Knowledge on Physics are Required**

To achieve promising performance on SR problems for physics formulas, ParFam requires prior knowledge of all possible physics formulas, as from SRBench and the Cambridge Handbook of Physics Formulation. I think this prior knowledge is very strong and only specific to physics formulas, and is hard to be generalized for other SR problems in real-world applications.

**3. DL-ParFam**

The idea of DL-ParFam, a deep learning-based pretrain model for ParFam, is interesting. But it is currently more like a toy prototype, and only tested on very simple synthetic problems. To truly show the advantage of DL-ParFam over other pre-training-based SR methods, a concrete model design on real-world SR applications is required.

In DL-ParFam, the model only takes the function value y as input to predict the mask c for all parameters, and all information of the function input x is completely ignored. It is hard to believe this approach can provide a reasonably good prediction for real-world SR applications, especially those with complicated structures.

To build the pre-trained model, DL-ParFam requires the input data x to have the same dimension $m$, and the data should be sampled on the same grid across all different data sets. Can this requirement be easily satisfied for physical SR problems and other SR problems?

**4. Experiments**

Since ParFam has a strong prior knowledge of the physical formulas, it is expected it can have promising performance on the physics SR problems. Indeed, according to the results, ParFam even discards part of the observed data, and only requires a subset of 500-1000 data points for coefficient optimization. It is hard to imagine this procedure could work well for real-world SR problems.

DL-ParFam is only tested on very simple synthetic problems. It is hard to judge its potential for solving real-world application problems with complicated structures.

**Questions:**

Please see the weaknesses section. I am willing to adjust my rating if the issues in weaknesses are well addressed.

[1] Symbolic Regression is NP-hard. TMLR 2022.

---

> ### Author Response · Authors · 2023-11-20
>
> We thank Reviewer LMH9 for their time and thoughtful comments on our paper and a chance to address their concerns.
> In the general response, we already addressed your questions and concerns regarding the question of whether ParFam is still symbolic regression, the physical prior knowledge, and the limitations of DL-ParFam. Hence, we only comment shortly on these here. Please also note the related adaptions to our paper that we mentioned in the general response.
>
> __Is It still Symbolic Regression?__
>
> As argued in the general response, from our point of view, ParFam is still an SR method and just combines the two steps of learning the function structure and optimizing the coefficients in the global search method.
>
>
> __Strong Prior Knowledge on Physics are Required__
>
> We agree that the interest of SR is not limited to physics and, thus, the scope of SR algorithms should also cover other fields. As argued in the general response, the chosen parametric family covers also a lot of forumlas from other areas. Moreover, ParFam is highly expressive and can hence approximate many other formulas well as can be seen by the achieved high accuracy.
>
> __DL-ParFam__
>
> We completely agree that it is not reasonable to omit the input data $x$ completely and to assume that all the data has the same size and dimension and is sampled on the same grid. This was a simplifying assumption for this paper to present the potential achievable by DL-ParFam. We hope to clarify the role of DL-ParFam in our general response.
>
> __Experiments__
>
> It is an interesting question if the strong physics prior is the reason why we were able to discard such a large amount of the data for the Feynman problems.
> However, note that also SRBench discards a lot of the data, since the original Feynman data set was extremely big ($10^5$) for these low dimensional problems (less than 10 dimensions), probably because the authors of the Feynman data set, Udrescu and Tegmark [1], had to train NNs on the data to test multiple symmetries. Furthermore, the data set size we used is closer to the usual data set size used in symbolic regression: Strogatz data set (400 data points), Nguyen (20 - 100 data points), Livermore (20-1000), R-Rationals (20). Note that interestingly we observed now when comparing ParFam to "Symbolic physics learner: Discovering governing equations via Monte Carlo tree search" [2], see the reply to Reviewer 8mnU, that ParFam has not enough prior to find the desired function on the small standard ranges as specified in the Nguyen data set. Instead, it finds other simple functions which are a near-perfect approximation on the data domain, indicating that the prior in ParFam might not be stronger than in other methods. An example for this is the target function $\sin(x^2)\cos(x)-1$ on the domain $[-1,1]$, where it instead finds the function $0.569x^2 - 0.742 \sin(1.241x^2 - 2.059) - 1.655$ which is almost indistinguishable between $[-1,1]$.
>
>
> We thank Reviewer 5PDr again and hope that we were able to address the concerns appropriately.
>
> [1] Silviu-Marian Udrescu and Max Tegmark. AI Feynman: A physics-inspired method for symbolic regression. Science Advances, 6(16):eaay2631, 2020. doi: doi:10.1126/sciadv.aay2631.
>
> [2] Sun F, Liu Y, Wang JX, Sun H. Symbolic physics learner: Discovering governing equations via monte carlo tree search. ICLR. 2023. URL https://openreview.net/forum?id=ZTK3SefE8_Z.

---

> > ### Comment · Reviewer_LMH9 · 2023-11-21
> > **Thank you for the thorough response**
> >
> > Thank you for the thorough response. I've also read other reviewers' comments and the corresponding responses. Since some of my concerns have been appropriately addressed, I raise my score to 5.
> >
> > However, I also believe the contribution of ParFam alone is not enough for a clear acceptance, while a well-developed DL-ParFam could significantly strengthen the quality of the current work.

---

### Official Review · Reviewer_5PDr · 2023-11-02

**Soundness:** 2 fair
**Presentation:** 3 good
**Contribution:** 2 fair
**Rating:** 5
**Confidence:** 4

**Summary:**

This paper proposes a method for symbolic regression in which the purpose is to search for a mathematical formula describing given data. The proposed method, termed ParFam, defines a structure of the target equations in advance and then optimizes the coefficients using gradient-based methods. Owing to this problem transformation, the symbolic regression problem becomes a contiguous problem from a discrete one. In addition, the technique that combines ParFam and the neural network-based structure prediction of the sparsity of coefficients is introduced. The authors experimentally evaluate the performance of ParFam using SRBench and show that it can achieve state-of-the-art performance compared to other symbolic regression methods.

**Strengths:**

- A novel method for symbolic regression is presented, which translates the original discrete combinatorial optimization problem into the continuous optimization problem with a pre-defined structure of equations.
- The proposed ParFam achieves state-of-the-art performance on SRBench.
- This paper is well-written and easy to follow.

**Weaknesses:**

- When the number of the input variables of the equation increases, it seems hard for ParFam to handle the exponential growth of the number of parameters, as the authors describe in Section 4.
- As the authors stated in the introduction, symbolic regression aims to find a symbolic model with as few assumptions as possible. However, in ParFam, the form (structure) of the target equations is pre-defined by users. If the structure of the equation is not suitable for a given data, it cannot represent an appropriate equation.
- The experimental evaluation of DL-ParFam is limited to the synthetic datasets.

**Questions:**

- The concept of the proposed ParFam is somewhat similar to equation learner (EQL). Given an appropriate network architecture that corresponds to the equation structure of ParFam, the search space of EQL could be almost the same as ParFam. Could you describe the main difference and advantage of ParFam against EQL? Also, is there any experimental comparison of EQL and ParFam?
- Why is it difficult to apply and evaluate the DL-ParFam to SRBench?
- How is the sensitivity of the performance of ParFam for the regularization hyperparameter $\lambda$?
- Could you report the exact number of parameters to be optimized in ParFam?

---

> ### Author Response · Authors · 2023-11-20
>
> We thank Reviewer 5PDr for their time and thoughtful comments on our paper. In the general response, we already addressed your questions and concerns regarding the expressivity of the parametric family, the limited experimental evaluation of DL-ParFam, and the similarity of EQL and ParFam. Please also note the related adaptions to our paper that we mentioned in the general response.
> To answer the question shortly, why the prototype version of DL-ParFam cannot be used on the SRBench data set: These data sets require the pre-trained neural network to handle varying data set sizes and dimensions, which the simple prototype using a fully connected neural network is not able to. Thank you for pointing this out, we added this explanation at the beginning of Section 3.2:
>
> > Due to the prototype status of DL-ParFam, the ability to evaluate it on complex data sets, such as the Feynman dataset, is limited as the data to be processed is not sampled on the same grid. Therefore, we use synthetic data sets.
>
> Regarding the experiments with EQL, we will update our paper soon to show these results as well, thank you again for the idea to incorporate this. Regarding the exponential growth of the number of parameters, we agree that is a serious limitation, which is one of our main motivations to focus on DL-ParFam to reduce the number of parameters in the future.
>
> __How is the sensitivity of the performance of ParFam for the regularization hyperparameter $\lambda$?__
>
> Thank you for this very interesting question. Note that the role of $\lambda$ can be well understood, if the MSE in the loss function in Equation (2) is scaled by the norm of $y$, i.e., if we consider the relative error, which is done in our implementation.
>
> In this case, $\lambda$ depicts a quantifiable relation between good accuracy and sparse/small coefficients. This explains why $\lambda$ should probably be chosen to be less than 0.1 since otherwise there is a high probability that a model with very small coefficients exists, which has a lower loss than the true model.
> We occasionally also observed this for 0.01 in toy experiments while debugging ParFam and found that $\lambda = 0.001$ worked the best quite reliably. For this reason, we started the experiments on SRBench directly with that value. However, we think that a sensitivity analysis of this hyper-parameter would be a nice addition to our paper, which is why we ran additional experiments and added the results to the supplementary material, Appendix H. Interestingly, this shows that the performance of ParFam is not very sensitive to $\lambda$.
>
> __Could you report the exact number of parameters to be optimized in ParFam?__
>
> We only state the approximate number "a few hundred" in the numerical section, since the number of parameters depends strongly on the number of input variables and polynomial degrees. For example, if there are 3 input variables the maximal model parameter choices for the SRBench experiments yield a parametric family with 102 parameters, while if there are 5 input variables this results in a family with 296 parameters, and for 7 input variables 671 parameters. However, as described in Appendix E, there are multiple different model parameters, we iterate through, resulting in different parametric families with different numbers of parameters for each. We hope that this gives an idea of the number of parameters optimized for in ParFam.
>
>
> We again thank Reviewer 5PDr for their time and hope that we could answer their questions and concerns.

---

### Author Response · Authors · 2023-11-20
**General response (1/3)**

We thank the reviewers for their thoughtful and constructive review of our manuscript and apologize for the late response. We are happy to see that the general idea of ParFam and its strong performance on SRBench have been well received. In this comment, we would like to address the main concerns and questions raised by multiple reviewers:

__1. How restrictive is the choice of the paramtric family?__

 __1a. Is it still Symbolic Regression?__ (Reviewer LMH9)
This is a very interesting question, that we are happy to address. For the following reasons, we believe that ParFam qualifies as a symbolic regression method: The subdivision of Symbolic Regression into finding the analytic expression and then finding the parameters is the most common approach to SR, not the definition of SR in general. The goal of SR is to find a symbolic/interpretable formula to approximate given data while trying to keep the assumptions on the formula to a minimum. One may observe that ParFam does not really omit the first step of finding the analytical expression, instead, it performs both steps at the same time since the optimizer will set (if the method is successful) most of the parameters to 0, giving rise to the actual analytic expression. This is the reason, why we claim that we translated the discrete optimization problem of SR (here we refer to actually finding the analytic expression) to a continuous one since now the expression can be found by using continuous optimization. Furthermore, note that the different analytic formulas are not trivially connected as can be seen by looking at the formulas correctly identified by ParFam from the SRBench data sets, e.g., $mom \sqrt{Bx^2+By^2+Bz^2}$, $2U(1-\cos(kd))$ and $\exp(-((\theta-\theta_1)/\sigma)^2/2)/(\sqrt{2\pi} \sigma)$. As we believe that this is a very valuable question, we added the following paragraph in the introduction:

> Most SR algorithms approach the problem by first searching for the analytic form of $f$ and then optimizing the resulting coefficients. In contrast, only a few algorithms follow the same idea as ParFam, to merge these steps into one by spanning the search space using an expressive parametric model and searching for sparse coefficients that simultaneously yield the analytical function and its coefficients.

 __1b. As the authors stated in the introduction, symbolic regression aims to find a symbolic model with as few assumptions as possible. However, in ParFam, the form (structure) of the target equations is pre-defined by users. If the structure of the equation is not suitable for a given data, it cannot represent an appropriate equation.__ (Reviewer 5PDr) and __Strong Prior Knowledge on Physics are Required__ (Reviewer LMH9)

This is a very important point which we also mention as one of the limitations in the paper. However, we aimed to define a very expressive family of functions, which covers most of the relevant formulas (see the Cambridge Physics Handbook for example).

Due to the strong interest of the physics community in SR and also the focus of SR community on physical problems (cf. SRBench), we also focused on physical problems.
However, we agree that the interest of SR is not limited to physics and, thus, the scope of SR algorithms should also cover other fields.
Fortunately, the structure of the parametric family is highly expressive while producing formulas that are interpretable. In Symbolic Regression, it is not simply the goal to find the one correct function, but it is often enough to find a simple function that approximates the relation between x and y sufficiently well. The high expressivity of ParFam can be seen by the high accuracy solution rate (>90%) of ParFam on the SRBench data, even though roughly a third of the data sets were not covered by our choice for the polynomial degree and activation functions for time reasons. ParFam also favors more interpretable formulas since the composite function it omits is not just often unreasonable, but also hard to interpret (e.g., $\sin(\cos(x))$).
Moreover, many of the functions we omit, are also omitted by other methods like DSR [3] and Discovering governing equations via Monte Carlo tree search [4], since they claim $\sin(\cos(x))$ to be not a reasonable function. The second reason for this structure is that it is highly expressive while producing formulas that are more interpretable. To clarify on this topic we added the following sentence:

> In addition to its high relevance to known physical laws, the structure of ParFam is chosen due to its inherent interpretability since it avoids complicated compositions and its high expressivity even if the true formula cannot be recovered, as can be seen in our experiments in Section 3.1.

---

> ### Author Response · Authors · 2023-11-20
> **General response (2/3)**
>
> __2. The evaluation of DL-ParFam on synthetic data is not convincing enough to introduce a new algorithm that should be able to handle real-world data__ (Reviewer 5PDr, LMH9, and 1Ff3):
>
>    We agree with the valid concern of the reviewers regarding the use of synthetic data for DL-ParFam evaluation. We acknowledge the importance of demonstrating applicability to real-world data and agree that our current evaluation does not fulfill this criterion. However, it is crucial to emphasize that this was not the goal of our paper. Our primary objective was to introduce ParFam as a competitive SR method, with a secondary goal of showcasing its potential for efficient pre-training. To show this secondary goal, we opted for a prototype evaluated on synthetic data to show that this can improve on ParFam. The variability in data (e.g., data on variable grids, high-dimensional data, varying number of data points, varying base functions, etc.) makes the choice of the precise architecture, training procedure, data sampling scheme, and loss function for DL-ParFam very complicated. Since this is not directly related to ParFam, we outsourced this to future work.
>
> To ensure that this is clear and since we do not want to claim that DL-ParFam outperforms existing approaches on real-world data we added several clarifications in the paper. In the contributions section, we changed it to
>
> > Furthermore, we introduce a prototype of the extension DL-ParFam, which shows how the structure of ParFam allows for using a pre-trained NN, potentially overcoming the limitations of previous approaches.
>
> In Section 2.2 we added the first part of the following sentence
>
> > Since the main focus of this paper is ParFam and the particular choices are not directly related to ParFam, we opt for a straightforward implementation of DL-ParFam to demonstrate its effectiveness on synthetic data.
>
> Furthermore, in Section 3.2 we added at the beginning:
>
> > Due to the prototype status of DL-ParFam, the ability to evaluate it on complex data sets, such as the Feynman dataset, is limited as the data to be processed is not sampled on the same grid. Therefore, we use synthetic data sets.

---

> > ### Author Response · Authors · 2023-11-20
> > **General response (3/3)**
> >
> > __3. What is the main difference between ParFam and EQL__ (Reviewer 5PDr and 1Ff3) __and can we see a numerical comparison between those two methods__ (Reviewer 5PDr):
> >
> >    We appreciate the reviewers' interest in understanding the distinctions between ParFam and EQL, since we agree that this differentiation is an important part to justify the novelty of our approach. Your question has allowed us to delve deeper into these differences and provide a more nuanced comparison.
> >
> >    Let us start with the similarities. Both methods employ a parametric family (in the case of EQL as a shallow neural network) and aim to learn sparse parameters to represent the symbolic function. These parametric families, however, differ in some important aspects. While the EQL network consists of linear layers and non-linear activation functions, ParFam has rational functions as "layers" before and after the base/activation functions.
> >
> >    Since EQL represents polynomials using the multiplication node, it requires multiple hidden layers, since, e.g., one hidden layer can only represent polynomials of degree two. From our experience with similar architectures, these multiple layers make the loss landscape more complex and the resulting functions harder to interpret since they can contain compositions of trigonometric functions, for example. The original paper [1] uses as base functions only the sigmoid, sin, and cos. We are not sure why this restriction is necessary, but the exponential function might probably cause numerical problems in this compositional setting, and for the logarithm and square root they mention: "Further nonlinearities, such as (square) roots and logarithms, could in principle be useful for learning physical equations, but they pose problems because their domains of definition is restricted to positive inputs." [1] In the follow-up EQL paper [2], they introduce the division as an activation function with two inputs, but only in the last layer, which makes expressions with divisions within unary base functions like $\sin(x_1/x_2)$ impossible. A further difference between EQL and ParFam is that we aimed at minimizing the number of parameters by avoiding non-unique parameterizations, which allows the use of global optimizers (without incredibly increasing computational costs).
> > The restrictions in the expressivity of EQL (probably) caused other papers (e.g. DSR [3]) to not use EQL as a baseline in their experiments and it complicates the evaluation of EQL on SRBench since many of the functions there cannot be expressed by EQL. However, we admit that it is an interesting question to compare EQL to ParFam, which is why we are currently working on running EQL on a reduced version of SRBench, which omits all functions that EQL does not cover. The results will be added as supplementary material. Moreover, we added some more details concerning the relation of ParFam and EQL in the related work paragraph in the introduction.
> >
> > We again want to thank all reviewers for their time and valuable feedback and hope that we were able to clarify the main questions and concerns.
> >
> > [1] Georg Martius and Christoph H. Lampert. Extrapolation and learning equations. In 5th International Conference on Learning Representations, Workshop Track Proceedings. OpenReview.net, 2017. URL https://openreview.net/forum?id=BkgRp0FYe.
> >
> > [2] Subham Sahoo, Christoph Lampert, and Georg Martius. Learning equations for extrapolation and control. In Proceedings of the 35th International Conference on Machine Learning, volume 80 of Proceedings of Machine Learning Research, pp. 4439–4447. PMLR, 2018. URL http://proceedings.mlr.press/v80/sahoo18a.html.
> >
> > [3] Brenden K. Petersen, Mikel Landajuela, T. Nathan Mundhenk, Cl´audio Prata Santiago, Sookyung Kim, and Joanne Taery Kim. Deep symbolic regression: Recovering mathematical expressions from data via risk-seeking policy gradients. In 9th International Conference on Learning Representations, ICLR 2021. OpenReview.net, 2021. URL https://openreview.net/forum?id=m5Qsh0kBQG.
> >
> > [4] Sun F, Liu Y, Wang JX, Sun H. Symbolic physics learner: Discovering governing equations via monte carlo tree search. ICLR. 2023. URL https://openreview.net/forum?id=ZTK3SefE8_Z.

---

> > > ### Author Response · Authors · 2023-11-21
> > > **Update Paper: Comparisons with EQL and SPL**
> > >
> > > Please note that we uploaded a new revision of the paper in which we added the requested comparisons with EQL [1,2] and SPL [3] as Appendices I and J. The comparison with EQL in Appendix I shows the strong advantage of ParFam compared with EQL. From the experiments on the Nguyen benchmark in Appendix J, it can be seen that ParFam is competitive with SPL  and NGGP [4] even on more artificial, non-physics-inspired benchmarks.
> > >
> > > [1] Georg Martius and Christoph H. Lampert. Extrapolation and learning equations. In 5th International Conference on Learning Representations, Workshop Track Proceedings. OpenReview.net, 2017. URL https://openreview.net/forum?id=BkgRp0FYe.
> > >
> > > [2] Subham Sahoo, Christoph Lampert, and Georg Martius. Learning equations for extrapolation and control. In Proceedings of the 35th International Conference on Machine Learning, volume 80 of Proceedings of Machine Learning Research, pp. 4439–4447. PMLR, 2018. URL http://proceedings.mlr.press/v80/sahoo18a.html.
> > >
> > > [3] Sun F, Liu Y, Wang JX, Sun H. Symbolic physics learner: Discovering governing equations via monte carlo tree search. ICLR. 2023. URL https://openreview.net/forum?id=ZTK3SefE8_Z.
> > >
> > > [4] T. Nathan Mundhenk, Mikel Landajuela, Ruben Glatt, Cl´audio P. Santiago, Daniel M. Faissol, and
> > > Brenden K. Petersen. Symbolic regression via deep reinforcement learning enhanced genetic
> > > programming seeding. In Advances in Neural Information Processing Systems, volume 34, pp.
> > > 24912–24923, 2021. URL https://proceedings.neurips.cc/paper/2021.

---

### Meta-Review · Area_Chair_WBBA · 2023-12-05

**Metareview:**

This paper investigates the optimization phase over continuous coefficients, *after* a symbolic form has already been determined. This continuous optimization problem becomes a regular MSE-regression problem, with possibly a LASSO regularization as a sparsity constraint (to zero-out some of the symbolic components). Evaluations are performed over physics-specific regression tasks.

The authors claim that regular optimizers (BFGS, gradient descent) are insufficient in this scenario, and instead propose a "basin-hopping" method. Following this, the authors then propose a deep learning variant, in which a pretrained MLP first predicts the sparsity pattern.

The first core issue is the misleading title - it would be better if the paper had been written to propose the problem of optimizing continuous coefficients post-symbolic regression, which is a fine and interesting topic to investigate.

Then the responsibilities of the paper are:
 * To explain why physics-based symbolic regression problems are the only ones worth studying.
* To show that regular coefficient optimizers are suboptimal, and why this is the case.
* Why the new proposed method (ParFam) should work better, and experimentally validate this claim.
* Why DL-ParFam is even better, and also experimentally validate this claim.

Unfortunately, the paper falls short of satisfying these responsibilities. It is unclear at all why regular coefficient optimizers are suboptimal. The only experimental evidence can be seen in Appendix B where the proposed basin-hopping outperforms other methods, but more evidence needs to be shown (e.g. does increasing continuous dimensionality make optimization more difficult for baselines?).  As numerous reviewers have also stated, DL-ParFam is also only validated on synthetic toy datasets, and it is unclear how it performs on real-world datasets.

Thus the current recommendation is reject, and I strongly suggest that the authors rephase their work.

**Justification For Why Not Higher Score:**

There were core issues that could not be resolved just from rebuttal phase:
* Misleading title / representation
* Poor experimental evidence + lack of comprehensive comparisons against other optimizers
* Over-focus on physics-based equations

**Justification For Why Not Lower Score:**

N/A

---

### Decision · Program_Chairs · 2024-01-16

Reject